# Twitter Mining for Detecting Interest Trends on Biodiversity: Messages from Seven Language Communities

**Shu Ishida [1], Takanori Matsui [1],\*, Chihiro Haga [1], Keiko Hori [2], Shizuka Hashimoto [3] and Osamu Saito [4]**

1. Graduate School of Engineering, Osaka University, Yamadaoka 2-1, Suita 565-0871, Osaka, Japan
2. Institute for Future Initiatives, The University of Tokyo, 1-1-1 Yayoi, Bunkyo-ku, Tokyo 113-8657, Japan; hori.k@ses.usp.ac.jp
3. Department of Ecosystem Studies, The University of Tokyo, 1-1-1 Yayoi, Bunkyo-ku, Tokyo 113-8657, Japan
4. Institute for Global Environmental Strategies, 2108-11, Kamiyamaguchi, Hayama 240-0115, Kanagawa, Japan
\* Correspondence: matsui@see.eng.osaka-u.ac.jp; Tel.: +81-06-6879-7407

**Abstract:** The recent rates of global change in nature are unprecedented in human history. The Intergovernmental Science-Policy Platform on Biodiversity and Ecosystem Services (IPBES) has proposed a framework to achieve transformative change. Transformative change with respect to nature will be driven by recognizing the values people have; making inclusive decisions based on these values; restructuring policies, rights, and regulations in accordance with them; and transforming social norms and goals that can drive change. Social media is a new source of information and a modern tool for monitoring public opinion on human–nature interactions. This study identified commonalities among seven language communities (the six official languages of the United Nations and the Japanese language), demonstrating the uniqueness of the Japanese community by comparing hashtags in tweets that include the term biodiversity and determining differences in interest and concern about biodiversity from the past to the present. Tweets accessible at the end of 2021 that focus on biodiversity were collected from the Twitter server and used to form a text dataset. Interest was then qualitatively and quantitatively identified using natural language processing technology. Engagements and diversity indices were found to be on the rise in all language communities. We found that the Japanese language community has a different perspective on the relationship between biodiversity and humans from the scope of the IPBES conceptual framework. Future work should examine the relationship between passion for biodiversity and the Sustainable Development Goals. In addition, collaboration with various people around the world is necessary to understand the concept of biodiversity in different traditions and cultures.

**Keywords:** biodiversity; social media; Twitter; natural language processing; data mining

## 1. Introduction

The Intergovernmental Science-Policy Platform on Biodiversity and Ecosystem Services (IPBES) noted that the rate of recent global change in nature is unprecedented in human history and proposed a framework for transformative change [1]. According to this framework, the direct drivers of change in nature with the greatest impact globally are changes in land and sea use, direct exploitation of organisms, climate change, pollution, and invasion of alien species. These direct factors arise from a set of underlying factors called indirect drivers that are supported by social values and behaviors, including production and consumption patterns, human population dynamics and trends, trade, technological innovations, and both local and global governance. Therefore, to achieve transformative change from the current trend to a more sustainable one, it is necessary to recognize effective points of intervention across values and behaviors (both indirect and direct factors) and governance in collaboration with various stakeholders [2].

Transformative change entails a fundamental, system-wide reorganization across technological, economic, and social factors, including paradigms, goals, and values [1]. The context of transformative change differs according to history, culture, and social institutions (Figure 1). From the socioecological perspective, transformative change depends on eight leverage points and five collaborative implementations of priority governance interventions, termed levers [3]. As demonstrated in a set of values-centered leverage points in [2], "shifting societal norms and goals" will have the most significant impact. In addition, recognizing the nature values people have; making inclusive decisions based on these values; restructuring policies, rights, and regulations; and transforming social norms and goals can drive transformative change. Goal 1 of the Aichi Targets established in 2010 by the Convention on Biological Diversity (CBD) [4,5] has as its first aim that "people are aware of the values of biodiversity and the steps they can take to conserve and use it sustainably". In response to the National Biodiversity Strategies and Action Plans mandated by the CBD [6], the Japanese government has also set the following: "Strategy (1): Letting biodiversity sink in Japanese society" [7]. Further, the Kunming-Montreal Global Biodiversity Framework also proposed enhancing communication, education, and awareness of biodiversity and the uptake of this framework by all actors in Section K (outreach, awareness, and uptake) [8]. Determining the leverage points is crucial for considering viewpoints that will stimulate people's interest. Thus, the Japanese government had developed the 6th National Biodiversity Strategies and Action Plan and setting "awareness and action for biodiversity in the consumption and production" as the main pillar of the strategy [9,10].

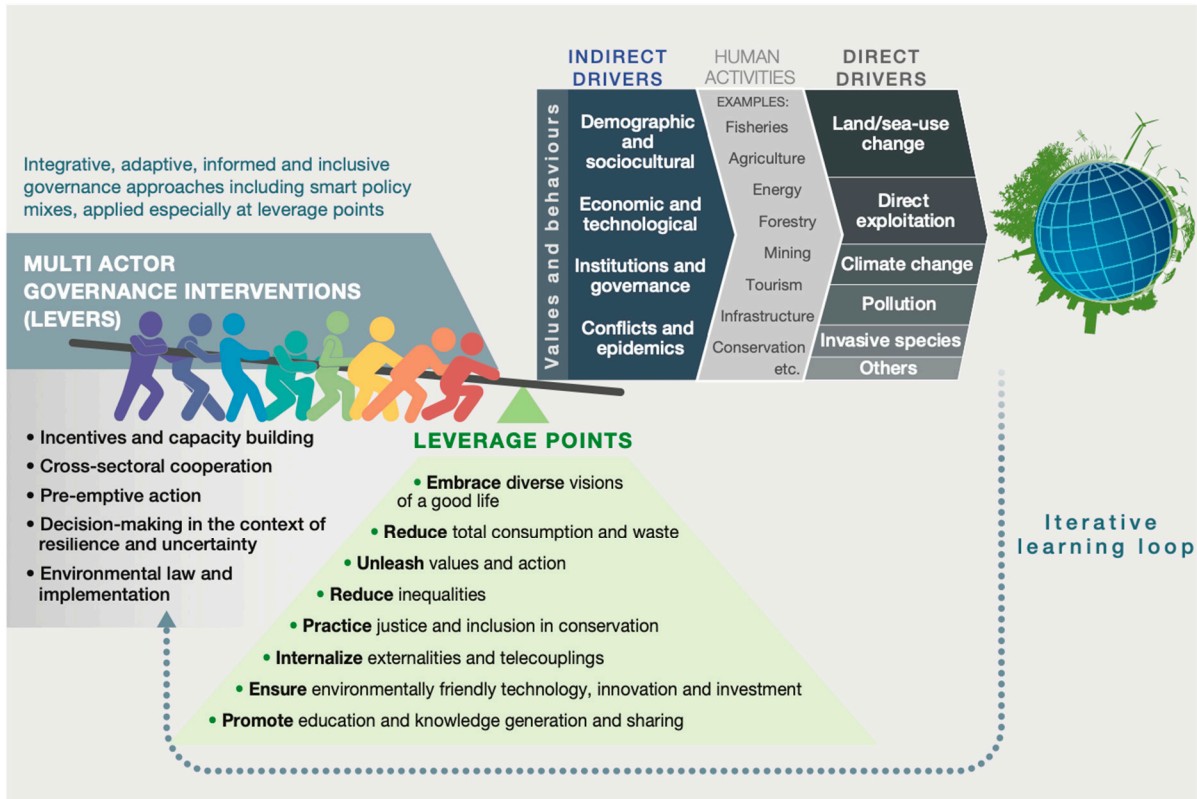

**Figure 1.** Transformative change in global sustainability pathways (IPBES, 2019) [1]. This image depicts the connected processes from the cultural aspect, indirect drivers, human activities, and direct drivers of ecosystems, as well as eight leverage points that are essential for transformation and five levers to enable transformative change. License: Creative Commons Attribution 4.0 International (CC BY 4.0).

In this trend, the CBD assessed the achievement of Goal 1 of the 10-year Aichi Targets [4,6]. The Ministry of the Environment of Japan checked time-series trends in the interest of nature and engagement in biodiversity conservation from the 1990s to the 2010s [11]. However, these were limited questionnaire surveys with a limited number of respondents, and they did not aim to understand the biodiversity values people have. The CBD indicated that there was no globally consistent information identifying trends in awareness or willingness to act on biodiversity [5]. Leadley et al. (2022) indicated the importance of a monitoring system that can discover and identify the drivers that cause changes in biodiversity, the actions that lead to the expected biodiversity outcomes, and the need to establish a set of readily monitored predictive indicators to design proactive planning and actions [12].

Within this context, social media is a novel source of information and an up-to-date tool for monitoring public opinion in human–nature interactions [13,14]. Analysis of citizens' interests through social media can be implemented through various means, including smart city [15–17], traffic [18–20], and energy [21]. Several types of social media have been used for this purpose, and one of the most common is Twitter. Twitter is a service for friends, family, and coworkers to communicate and stay connected through the exchange of quick and frequent messages called tweets, which may contain photos, videos, links, and text [22]. Monetary Daily Active Usage/Users counted 237.8 million active users in 2022 [23]. Twitter features a service called the full-archive search for use in academic research. This allows people in academia to access the full corpus of Twitter data back to the first tweet posted in March 2006 [24]. It is expected that a Twitter search will help clarify people's interest in biodiversity.

A great deal of research on climate change has been conducted using Twitter. For instance, Pearce et al. (2014) investigated how people reacted to the Integrated Assessment Report of the IPCC [25], and Kim and Cooke (2018) observed speaking about climate change and ocean acidification when the United States withdrew from the Paris Agreement [26]. Molodtsova (2014) measured the influence of mass media on attitudes linking local extreme weather events to climate change in the United States [27]. Arlt et al. (2018) investigated the effects of media and interpersonal communication on participation in climate discourse online and revealed that receiving information on climate change from social media, including Twitter; active information seeking online; and interpersonal conversations strongly encourage participation [28]. Sanford et al. (2023) examined the influence of the emotional framing of messages posted by environmental activists on engagement and behavioral intentions on climate action [29]. Studies related to COVID-19 have included analyses of the impact of a global pandemic on mitigation behavior and information communication on climate change on Twitter [30–32]. As state-of-the-art research, Thakur (2023) combined Twitter mining and sentiment analysis and detected the difference in opinions via emotions [33].

Twitter has also been utilized in research on biodiversity. For example, Twitter helped in identifying endemic species [34] and invasive alien species [35], and social media data (including Twitter data) were compared to measure the popularity and number of visitors to a national park [36]. Jarić et al. (2016) assessed whether the use of Latin or vernacular names was more effective when data mining on the web [37], and Papworth et al. (2015) explored the transmission of conservation research through online news and social media [38]. Furthermore, several studies have covered people's interest in biodiversity conservation. Hawkins and Silver (2016) and Macdonald et al. (2017) analyzed discussions of and reactions to seal and lion hunting, respectively [39,40]. Hammond et al. (2022) conducted a content analysis of tweets about elephants [41]. Kidd et al. (2018) assessed the attention to threatened species on Twitter and found that many threatened species were not mentioned at all [42]. In an important study of biodiversity mainstreaming, Cooper (2019) investigated 22 biodiversity-related keywords in 31 languages in online newspapers, social media,

and internet searches to monitor Aichi Target 1 [43]. They also proposed a method to measure the progress of Aichi Target 1. Ohtani (2022) also attempted to reveal the emotional tendencies behind contexts in which biodiversity has been used on Twitter with sentiment analysis. The study presented changes in the discourse surrounding biodiversity in English tweets over the past 10 years and the possibility of developing quantitative methods such as natural language processing [44]. Barrios-O'Neill (2020), the most relevant study to our work, analyzed the advocacy of environmental NGOs on Twitter by screening advocacy relating to biodiversity and revealed that Twitter advocacy was dominated by climate change, overexploitation, and plastic pollution, but major threats to agriculture, urbanization, invasions, and pollution were rarely addressed [45].

It is important to clarify how people's interest in biodiversity has changed and whether it is currently designing the necessary levers to transform values and behaviors related to biodiversity. Menendez et al. (2018) analyzed tweets that used the hashtag #WorldEnvironmentDay on the respective day and identified interests related to the sustainable care of the environment and public health [46]. To deepen mainstreaming biodiversity among the public, our study aimed to identify the historical trends and specific interests of communication on biodiversity. In addition, we need to deeply dive into more specific concepts of the biodiversity domain by localizing the context to reflect the diversity of nature's values [2].

This study investigated the commonality among seven language communities (Japanese, which is the primary language of the authors, and six official languages of the United Nations) and the uniqueness of the Japanese community by comparing tweets that include the term biodiversity for 15 years. It also clarified changes in interest in and concern about biodiversity from the past to the present. This study formulated the following three research questions (RQs):

RQ1: Is the interest in biodiversity continuously activated (RQ1-1) and diversified (RQ1-2)? This result can contribute to monitoring the current progress of promoting people's interests on biodiversity.

RQ2: What are the shared interests among the language communities and the special interest of the Japanese language community? This can help share the diversity of the interests, depending on the indigenous communities.

RQ3: What promotes the interest in biodiversity in the Japanese language community? The knowledge can be used to support the effective measure design for enriching the people's values on biodiversity.

The detail of each RQ's interpretation and the methodology of the quantification are described in the next section.

## 2. Materials and Methods

As depicted in Figure 2, tweets on biodiversity were collected from the Twitter server to create the text dataset. After preprocessing the text dataset, interest in biodiversity was qualitatively and quantitatively identified via data mining using natural language processing technology. All programs in the pseudocode were implemented with Python (ver. 3.7.7).

| Input<br>**Data Collection** | method<br>**Analysis** | Output<br>**Evaluation** |

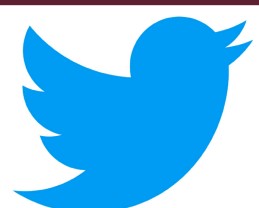

TWITTER, TWEET, RETWEET and the Twitter Bird logo are trademarks of Twitter Inc. or its affiliates

**Twitter Archive**
N = 5,140,273

**Period**
03/20(2006)-12/31(2021)

**Language**
Japanese
Arabic
Chinese
English
French
Russian
Spanish

**Preprocessing**

1. Extract texts
2. Extract hashtags by regular expression

**Data mining**

**Quantitative**

3. Create time series dataframe by language
4. Design metrics

**Qualitative**

5. Translate hashtags to English
6. Evaluate set operations of hashtags

**Quantitative**

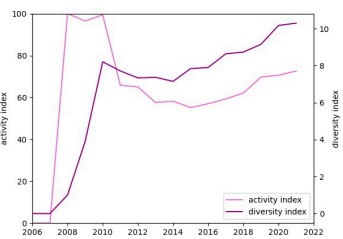

**Qualitative**

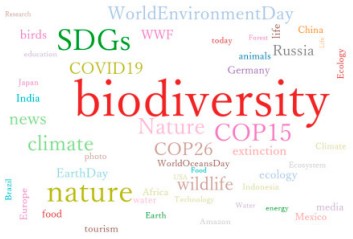

| line | code |
|---|---|
| 1 | `languages = [ja, en, es, fr, ru, cn, ar]` |
| 2 | `# tweet collection ————————————————` |
| 3 | `for language in languages:` |
| 4 | `        tweets = twitter_api_v2(query='biodiversity' in language)` |
| 5 | `        for tweet in tweets:` |
| 6 | `                hashtags_in_original_language = regular_expression(tweet)` |
| 7 | `                hashtags_in_english = googletrans(hashtags_in_original_language)` |
| 8 | `                # metrics evaluation ————————————————` |
| 9 | `                trend_metrics = engagement_calculator(hashtags_in_english)` |
| 10 | `                network_metrics = cooccurrence_calculator(hashtags_in_english)` |
| 11 | `# set operation ————————————————` |
| 12 | `for language in languages[—ja]:` |
| 13 | `        commonality = intersect(language, ja)` |
| 14 | `        uniqueness = complement(language, ja)` |
| 15 | `        comparison = wordcloud(commonality, uniqueness)` |

**Figure 2.** Overall structure from collecting tweets to analysis and pseudocode. The **upper part** of Figure 2 is the overall analytical flow. The input is the Twitter dataset, and the outputs are quantitative and qualitative assessments of biodiversity interests on Twitter. The **lower part** of Figure 2 is the pseudocode of the calculation process in the analysis, which is described in detail below.

### 2.1. Twitter Data Collection

Twitter API v2 [47] was used to collect tweets related to biodiversity through a data collection program. Japanese and the six official languages of the United Nations were selected as the target language communities. The full archive search function for academic was used, and the endpoint of "GET/2/tweets/search/all" was used for the Twitter search [48]. The *query* parameters for the search matching in the seven language communities were as follows: Japanese (生物多様性), Arabic (التنوع البيولوجي), Chinese (生物多样性), English (biodiversity), French (biodiversité), Russian (биоразнообразие), and Spanish (biodiversidad) [49]. The exact match was applied, and partial match and synonyms were not used for the query. The collected tweet.fields parameters were "created_at" and "public_metrics". The max_result pa-

rameter was 500. The search period was from 21 March 2006 to 31 December 2021. The start_time and end_time parameters were dynamically set according to the request response and the rate limit. The trials to connect the endpoint were iterated at 3.5 s intervals and stopped for 600 s when the iteration was exceeded to the limit rate. The data collection was performed twice. The first search period was from 21 March 2006 to 31 March 2021, and the second was from 1 April 2021 to 31 December 2021. It took two weeks in early May 2021 for the first collection and one week in February 2022 for the second collection. This dataset was built using the available tweets at this point. Therefore, this dataset did not include tweets that were deleted in the past and the deleted tweets could not be collected anymore so we had to be careful of sampling bias. All tweets were included in the dataset regardless of the user properties. The dataset could include the tweets generated by bots. Bots represent a serious threat to users, as they can launch large-scale attacks and manipulation campaigns, and removal technologies such as machine learning can be applied to detect them [50]; however, we decided to use all the data to research the effects of the bots.

### 2.2. Hashtag (#) Extraction

The hashtag (#) function is widely used in tweets. Twitter users can annotate keywords with a hashtag to emphasize interest. This hashtag can help users to find conversations on certain topics and bring their own tweets greater attention [51]. Thus, users' interests can be identified by analyzing the types of topics mentioned in the hashtags in tweets related to biodiversity. The hashtags were extracted from the original tweets using regular expression-matching operations. Regular expression is a sequence of characters that specifies a match pattern in text strings. Tokens represented #(\w+|[^ -~。-°]+) in regular expression matching were extracted as hashtags. Hashtags related to personal names, excluding those of celebrities, were removed to ensure privacy. To preserve the original intentions of the users, no preprocessing such as altering text casing was applied.

### 2.3. Evaluation of Engagements and Diversity Indices

Evaluation of engagement and diversity indices of hashtags were conducted to address RQ1 and determine whether there is continuous interest in biodiversity (RQ1-1) and whether it is diversified (RQ1-2).

### 2.3.1. Engagement

The engagements were evaluated according to the number of retweets, likes, and replies the tweets had [52]. On Twitter, users can retweet and like posts, denoting the endorsement of tweets [53]. *Tweet i*, *RTi*, *like i*, and *reply i* were the number of tweets retrieved, the total number of tweets retweeted, the total number of tweets liked, and the total number of tweets replied to in year *i*, respectively. Using these variables, *RT_engagement i*, *like_engagement i*, and *reply_engagement i* were calculated with Equations (1)–(3), respectively.

$$RT\_engagement_i = \frac{RT_i}{Tweet_i} \tag{1}$$

$$like\_engagement_i = \frac{like_i}{Tweet_i} \tag{2}$$

$$reply\_engagement_i = \frac{reply_i}{Tweet_i} \tag{3}$$

### 2.3.2. Diversity Index

Shannon entropy was used as the diversity index of interest in biodiversity to address RQ1-2. Shannon entropy is a metric used in information theory, and it is de-

fined as the amount of information in a bit unit [54]. Thus, if various topics (hashtags) emerge in year *i*—making topic (hashtag) distribution more divergent—then Shannon entropy is high. However, Shannon entropy is low if the topics (hashtags) are concentrated on a few specific interests.

The amount of information obtained for event *e* occurring with probability *p* is represented by Equation (4). Therefore, Shannon entropy determines probability distribution $p_1, p_2, p_3, \cdots, p_n$ of the proportion of hashtag *n* (*n* = 1 to *N*) appearing in year *i*, which was used as the diversity index of year *i* and is quantified by Equation (5).

$$I(e) = -log_2 p(e) \ [bit] \tag{4}$$

$$diversity \ index_i = -\sum_{n=1}^{N} p_n log_2 p_n \ [bit] \tag{5}$$

*2.4. Semantic Analysis of Hashtags*

The most common hashtags were visualized and compared between the Japanese community and the other six language communities to address RQ2, which investigated the shared interest among the language communities and the special interest of the Japanese language community, and RQ3, which regarded the type of matters that promote interest in biodiversity in the Japanese language community. Further, hashtags were used as nodes in the network analysis to calculate the mediating centrality of the hashtags with the highest number of hashtags in each language community. All languages were translated into English using Google Translate API (ver. 4.0.0rc1) [55] to set the operations of the hashtags. The translations to English were accepted without evaluating the accuracy of the translation due to limitations of linguistic and cultural understanding.

2.4.1. Popular Hashtags in Language Communities

The most commonly used hashtags that appeared in language community *l* were counted to identify interest in biodiversity during the entire sample period. The hashtags with the highest proportions were considered as the interest of the language community *l*.

2.4.2. Common and Unique Hashtags among Language Communities

The set operation was conducted to find hashtags used among the Japanese language community and the other six language communities to identify the commonality and uniqueness of interest in biodiversity within a language community. The set of hashtags that appeared in the Japanese language community and the other six language communities during the whole period was defined as $U_{japanese}$ and $U_{6langages}$, respectively. The intersection set between $U_{japanese}$ and $U_{6langages}$ was $U_{common}$, which represented the shared interest in biodiversity among the language communities. $U_{japannese\_only}$ and $U_{6langage\_only}$ were obtained using different sets of hashtags in each language community. These are presented in Equations (6)–(8).

$$U_{common} = \{U_{Japanese} \cap U_{6languages}\} \tag{6}$$

$$U_{japanese\_only} = \{hashtag | hashtag \in U_{japanese} \cap hashtag \notin U_{6languages}\} \tag{7}$$

$$U_{6languages\_only} = \{hashtag | hashtag \in U_{6languages} \cap hashtag \notin U_{japanese}\} \tag{8}$$

These hashtags were visualized in a word cloud [56], in which the size of each word represented the popularity of a given hashtag. The sizes in the different sets were adjusted using $H_l$ (*l* = *ja*, *ar*, *ch*, *en*, *fr*, *ru*, *es*), which was the proportion of the

number of particular hashtags in each language community *l*. The size of the intersection set was adjusted using $I_{tag}$, calculated as the harmonic mean of $H_l$ and expressed by Equation (9).

$$I_{tag} = \frac{n}{\frac{1}{n}\left(\frac{1}{H_{ja}} + \frac{1}{H_{ar}} + \cdots + \frac{1}{H_{es}}\right)} \tag{9}$$

### 2.4.3. Comparison with IPBES Conceptual Framework (IPBES-CF)

To compare interest in biodiversity, popular hashtags were compared with the IPBES-CF, developed by Díaz et al. (2015) [57] as depicted in Figure 3. The IPBES-CF is a highly simplified model of the complex interactions between the natural world and human societies. It comprises six interlinked elements—(1) nature, (2) nature's contributions to people (NCP), (3) anthropogenic assets, (4) institutions and governance systems and other indirect drivers of change, (5) direct drivers of change, and (6) good quality of life. The most commonly used hashtags in Japanese and English language communities were categorized according to these six elements, and their distributions were compared.

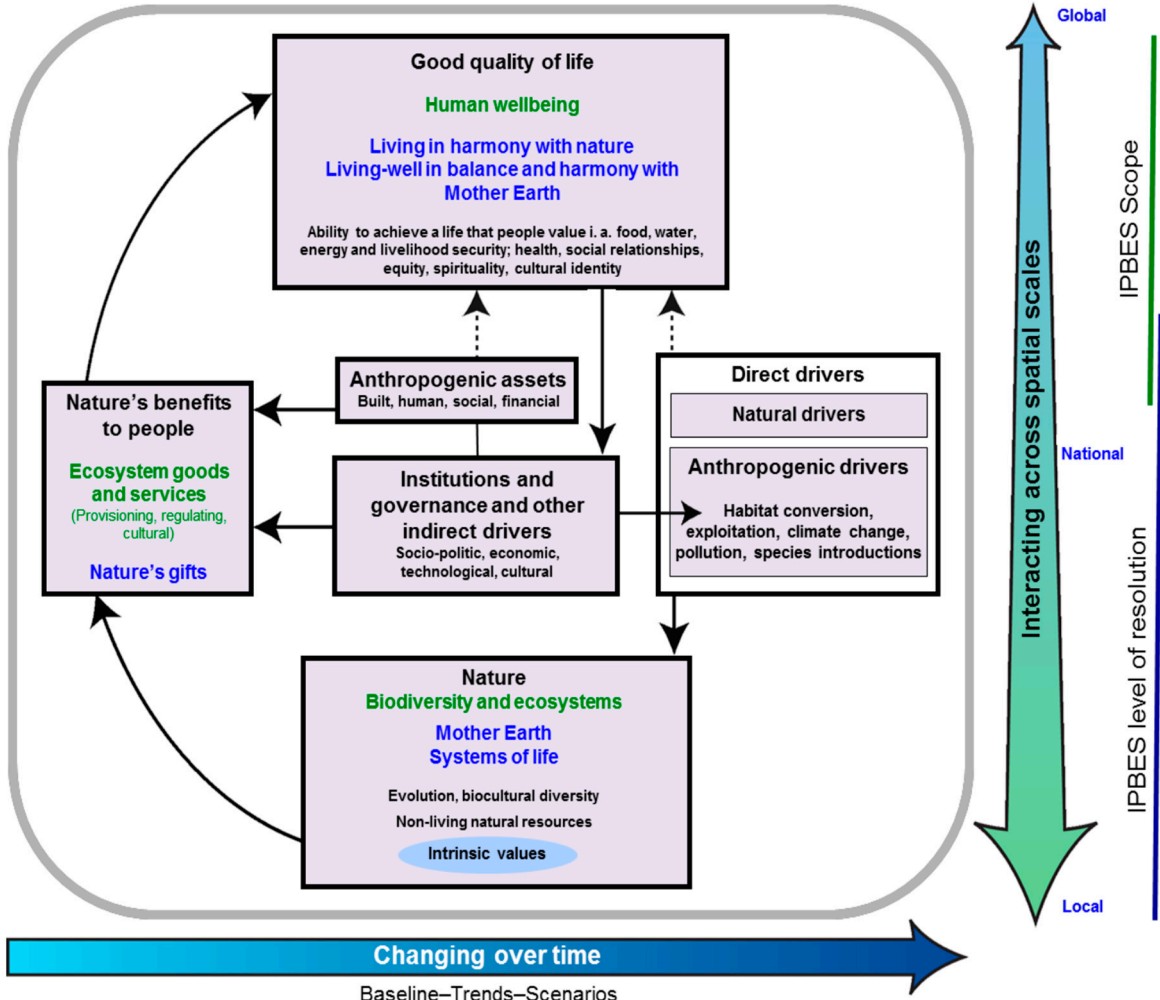

**Figure 3.** IPBES-CF [57]. Anthropogenic assets, institutions and governance, and other indirect drivers define the dynamics of direct drivers and the system of nature, nature's contribution to people, and good quality of life. The next institutions and governance and other indirect drivers are established based on the status of quality of life, which form multiple feedback loops of socioecological systems.

2.4.4. Network Analysis

To understand the context of the hashtags, the co-occurrence relationship of the hashtags was identified, and network graphs were created to calculate the betweenness centrality with the NetworkX (ver. 2.4) library [58], which is a metric of the importance of network nodes (Equation (10)). In addition, the betweenness centrality of the Japanese language community was calculated to identify the time-series trend of important concepts. $b_i$ denotes the betweenness centrality of node $i$; $\sigma_{st}(i)$ is the number of the shortest passes that include node $i$; and $\sigma_{st}$ is the number of the shortest passes between $s$ and $t$ ($1 \leq s, t \leq n \cap s, t \neq i$).

$$b_i = \sum_{s=1(s \neq i)}^{n} \sum_{t=1(t \neq i)}^{n} \frac{\sigma_{st}(i)}{\sigma_{st}} \tag{10}$$

## 3. Results

### 3.1. Tweet Statistics

Table 1 presents the number of posted tweets in each language that included the term "biodiversity" during the whole search period. The total number of related tweets in all languages was 5,140,273. There were 301,571 such tweets in Japanese, which was 6% of the total number of tweets. English (2,925,742, 53%) and Spanish (1,422,724, 26%) accounted for approximately 80% of the total tweets. Russian (3450, 0.6%) and Arabic (14,975, 0.2%) had the least proportions.

**Table 1.** Collected tweets according to language.

| Language | Search Word | Tweets |
|---|---|---|
| Japanese | 生物多様性 | 301,571 |
| Arabic | لتنوع البيولوجي | 14,975 |
| Chinese | 生物多样性 | 4971 |
| English | biodiversity | 2,925,742 |
| French | biodiversité | 466,840 |
| Russian | биоразнообразие | 3450 |
| Spanish | biodiversidad | 1,422,724 |

Note: the languages are presented in alphabetical order except for Japanese.

Figure 4 displays the time series of the number of tweets according to language. Overall, the number of tweets increased in all language communities. The Japanese, Arabic, Russian, and Chinese language communities experienced a burst of activity in certain years. For instance, in 2010, Japan hosted the Tenth Meeting of the Conference of the Parties (COP) of the CBD, which may have led to a surge in the data. International events may contribute to the number of tweets on biodiversity.

### 3.2. Quantitative Trends in Hashtags

#### 3.2.1. Hashtag Statistics

Table 2 presents the statistics of hashtags appearing in tweets. The Japanese language community used 0.27 hashtags per tweet, while the English language community used 1.69 hashtags and tended to annotate with various hashtags in a tweet. Certain language communities, especially the English language community, repeatedly used the same hashtags an average of 12.8 times, while in other language communities (such as the Chinese, and Russian communities), hashtags were reused on an average of only twice. Regarding tweets about biodiversity, the top 10% most used hashtags took up more than 80% of the total number of hashtags used in the English, French, and Spanish language communities such that topics and words on Twitter were widely shared and repeated within the language communities. The Chinese and

Russian language communities experienced an opposite tendency. Thus, the hashtags performed differently in different language communities.

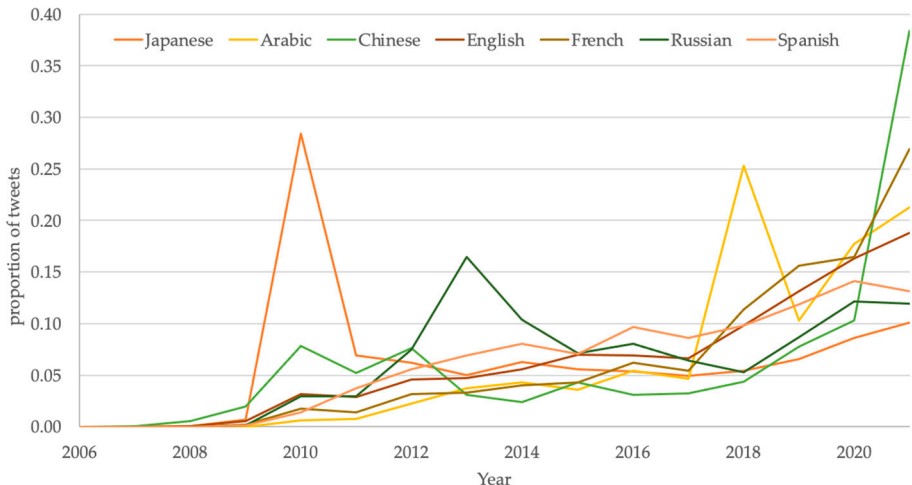

**Figure 4.** Time series of the number of related tweets according to language. The *x*-axis represents the year, and the *y*-axis represents the proportion of the number of tweets in each year to the total number of relevant tweets in a given language over the entire sample period.

**Table 2.** Statistics of hashtag frequencies.

| Language | Collected Tweets | Cumulative Number of Hashtags Used | Hashtags per Tweet | Cumulative Number of Tags Used/Number of Unique Hashtags | Occupied Proportion by | |
|---|---|---|---|---|---|---|
| | | | | | Top 10% Hashtags | Top 1% Hashtags |
| Japanese | 301,571 | 82,203 | 0.27 | 5.2 | 0.75 | 0.04 |
| Arabic | 14,975 | 16,278 | 1.09 | 3.9 | 0.66 | 0.10 |
| Chinese | 4971 | 1974 | 0.40 | 2.2 | 0.47 | 0.05 |
| English | 2,925,742 | 4,953,394 | 1.69 | 12.8 | 0.89 | 0.15 |
| French | 466,840 | 550,598 | 1.18 | 7.2 | 0.81 | 0.16 |
| Russian | 3450 | 2860 | 0.83 | 2.4 | 0.54 | 0.13 |
| Spanish | 1,422,724 | 1,423,735 | 1.00 | 7.4 | 0.81 | 0.09 |

3.2.2. Time series of Engagement and Diversity Index

The three engagements in each year are depicted in Figure 5a–c. The Japanese, Arabic, French, Russian, and Spanish language communities experienced increasing trends in RT_engagement from 2006 to about 2018–2020, after which it began to decline or slow. On the other hand, the Chinese and English language communities continued to see increases. Like_engagement was increasing in all language communities, but significant decreases were observed in Arabic, Chinese, and Russian language communities in 2021. Reply_engagement has generally been increasing but has been slowing for the Chinese, French, and Russian language communities in recent years. From the above, the English language community experienced a strong trend of increasing engagement as the number of tweets increased; the Japanese, French, and Spanish language communities experienced a gradual increase in the number of tweets and engagement; and the Arabic, Chinese, and Russian language communities experienced saturating or decreasing trends in engagements when the tweets increased. In contrast, the diversity index in Figure 5d depicts a monotonous increase in all language communities, indicating an increase in the diversity of interest.

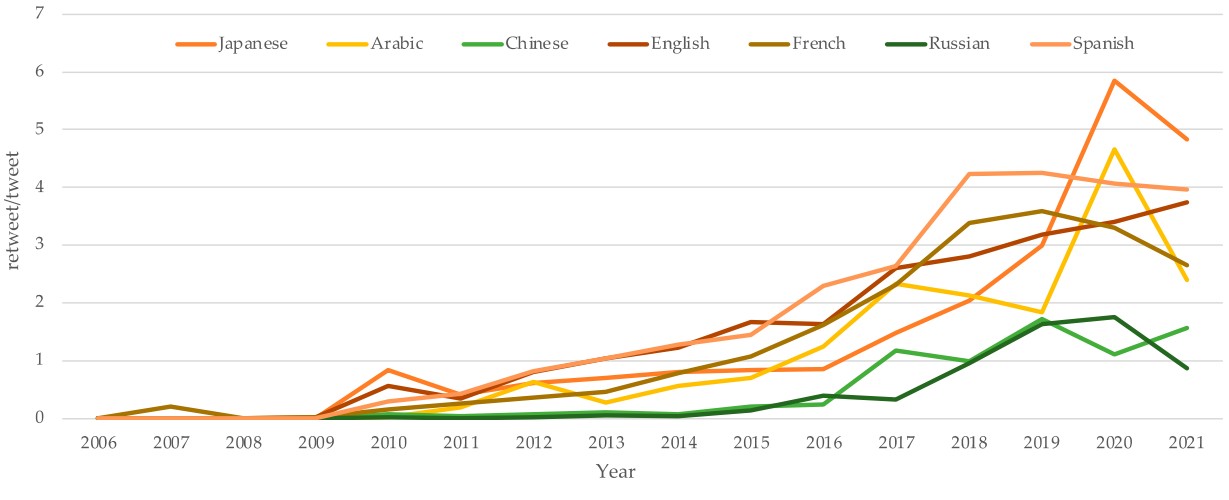

(**a**) RT engagement

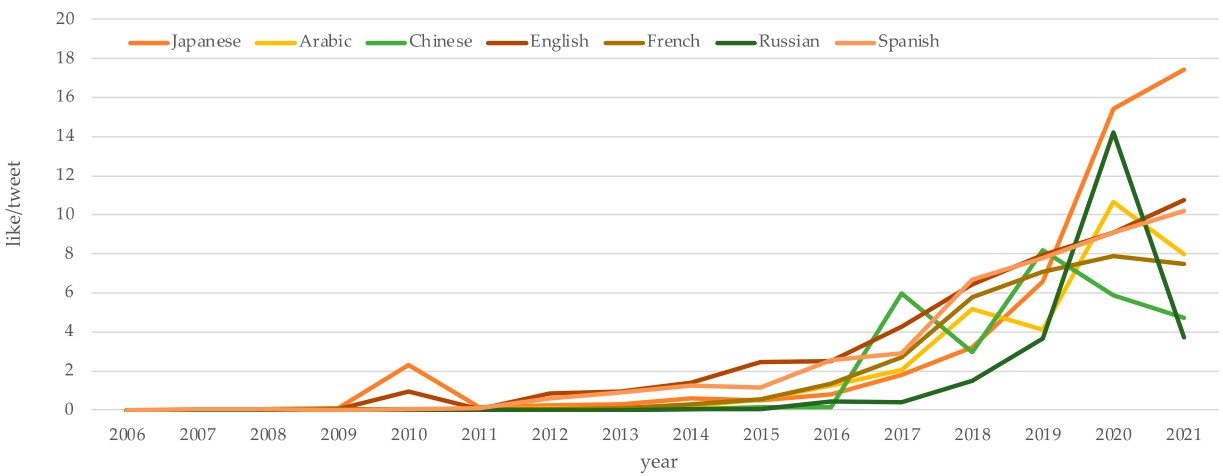

(**b**) Like engagement

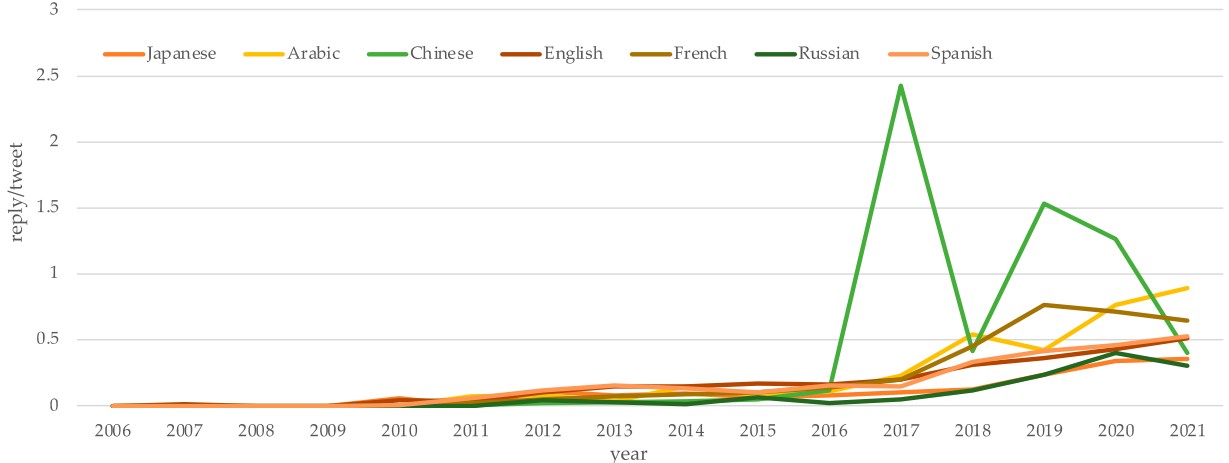

(**c**) Reply engagement

**Figure 5.** *Cont.*

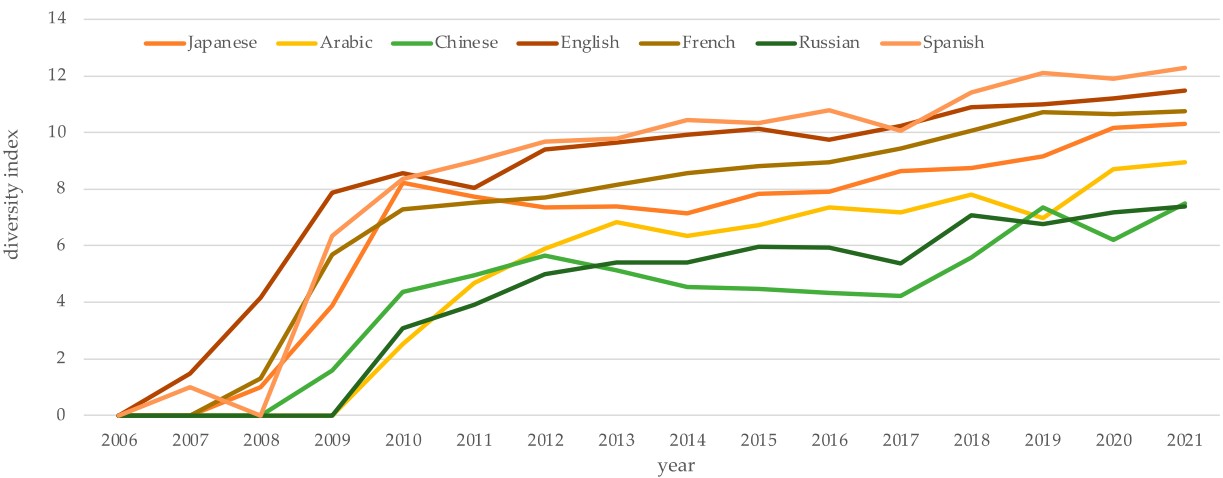

(**d**) Time series of the diversity index according to language

**Figure 5.** Time series of RT, like, reply, and diversity index according to language. An increase in (**a**–**c**) implies that the topics related to biodiversity were active, and an increase in (**d**) implies that the biodiversity interests were diversified.

### 3.3. Semantic Trends of Hashtags

### 3.3.1. Finding the Primary Interest

Table 3 reports the top 10 most frequently occurring hashtags for each language community presented in the original language and English as well as the proportion of the number to all tweets throughout the search period. All extracted hashtags from the dataset were split into seven language communities, and the most frequent 10 hashtags were extracted. The proportion of each hashtag was calculated by dividing the frequency by the total number of hashtags in each language. The hashtag #biodiversity itself ranked in the top 10 in all language communities. Between 16.02% and 9.12% of hashtags in the Arabic, English, French, Russian, and Spanish language communities included #biodiversity, implying that the word "biodiversity" itself signifies a common interest and concept perceived as a major topic for information dissemination across language communities. These language communities also tended to include the biodiversity-related concepts such as "nature", "wildlife", "ecology", "agriculture", "environment", "climate", and "planet". This tendency is very important to treat the biodiversity nexus issues such as the nature–climate nexus, so we will deepen the discussion later. However, the rate of use of #biodiversity in the Japanese and Chinese language communities was less than 2.60%. The word "biodiversity" itself may not be in common use and might be replaced with other words such as "richness of nature" in the Japanese and Chinese language communities. This terminology issue is also important for developing the strategies to expand and centralize the biodiversity concept in the societies. The CBD-COP made a strong contribution to promoting interest in biodiversity, so institutions and governance play a great role in managing nature's contribution to people, as shown in Figure 3. In addition, the interest in biodiversity was closely associated with the Sustainable Development Goals (SDGs), which include SDG Global Goals 14 (Ocean) and 15 (Land), in the Japanese language community compared with other language communities.

**Table 3.** Top 10 hashtags according to language.

| | Original | Translated | Tag/Cummulative Tags (%) | Original | Translated | Tag/Cummulative Tags (%) |
|---|---|---|---|---|---|---|
| | | Japanese | | | Arabic | |
| 1 | bdjp | BDJP | 3.77 | التنوع_البيولوجي | Biodiversity | 10.18 |
| 2 | cop10 | COP 10 | 2.88 | صديق_البيئة | environmentally friendly | 1.47 |
| 3 | 生物多様性 | Biological | 2.60 | موتمر_التنوع_البيولوجي | Conference of the biological diversity | 1.44 |
| 4 | biodiversity | biodiversity | 1.92 | السيسي | Sisi | 1.43 |
| 5 | SDGs | SDGs | 1.50 | الإمارات | UAE | 1.21 |
| 6 | エビフライ | Fried Shrimp | 1.37 | البيئة | The environment | 1.16 |
| 7 | かきあげ | Pounding | 1.33 | مصر | Egypt | 1.10 |
| 8 | ngo | ngo | 1.25 | أبوظبي | Abu Dhabi | 1.10 |
| 9 | npo | npo | 1.24 | قطر | Diameter | 0.92 |
| 10 | COP 10 | COP 10 | 1.21 | من_أجل_الطبيعة | From_Al_Al-nature | 0.88 |
| | | Chinese | | | English | |
| 1 | China | China | 4.56 | biodiversity | | 14.91 |
| 2 | COP 15 | COP 15 | 2.63 | Biodiversity | | 3.28 |
| 3 | 生物多样性 | Biological diversity | 2.03 | nature | | 2.27 |
| 4 | 中国 | China | 1.93 | conservation | | 1.22 |
| 5 | 新闻 | news | 1.67 | environment | | 0.93 |
| 6 | 世界 | world | 1.32 | climate change | | 0.89 |
| 7 | 西藏 | Tibet | 1.17 | wildlife | | 0.88 |
| 8 | Tibet | Tibet | 1.06 | photography | | 0.81 |
| 9 | 钓鱼岛 | Diaoyu Islands | 0.96 | 500pxrtg | | 0.77 |
| 10 | biodiversity | biodiversity | 0.91 | flora | | 0.77 |
| | | French | | | Russian | |
| 1 | biodiversité | biodiversity | 16.02 | биоразнообразие | biodiversity | 12.66 |
| 2 | biodiversite | biodiversity | 3.07 | экология | ecology | 2.90 |
| 3 | Biodiversité | Biodiversity | 2.29 | Биоразнообразие | Biodiversity | 2.69 |
| 4 | environnement | environment | 2.02 | biodiversity | biodiversity | 1.71 |
| 5 | nature | nature | 1.24 | ООН | UN | 1.57 |
| 6 | climat | climate | 0.98 | природа | Nature | 1.36 |
| 7 | Environnement | Environment | 0.63 | радиприроды | Radiriroda | 1.12 |
| 8 | ecologie | ecology | 0.48 | UN | UN | 1.01 |
| 9 | agriculture | agriculture | 0.46 | Socotra | Socotra | 1.01 |
| 10 | Nature | Nature | 0.41 | SaveSocotra | Savesocotra | 1.01 |

**Table 3.** *Cont.*

|  | Original | Translated | Tag/Cummulative Tags (%) | Original | Translated | Tag/Cummulative Tags (%) |
|---|---|---|---|---|---|---|
|  |  | Spanish |  |  |  |  |
| 1 | biodiversidad | biodiversity | 9.12 |  |  |  |
| 2 | Biodiversidad | Biodiversity | 5.92 |  |  |  |
| 3 | MedioAmbiente | Environment | 1.67 |  |  |  |
| 4 | medioambiente | environment | 1.09 |  |  |  |
| 5 | Planeta | Planet | 0.99 |  |  |  |
| 6 | naturaleza | nature | 0.67 |  |  |  |
| 7 | CambioClimático | Climate change | 0.49 |  |  |  |
| 8 | planeta | planet | 0.49 |  |  |  |
| 9 | Colombia | Colombia | 0.41 |  |  |  |
| 10 | México | Mexico | 0.39 |  |  |  |

Note 1: Retweets and special symbols were excluded. Note 2: The words in the translated column are hashtags as translated from each language into English by Google Translate [54]. However, the accuracies of the translations were not evaluated due to linguistic limitations.

### 3.3.2. Commonalities of Interest among Language Communities and the Uniqueness of the Japanese Language Community

The Venn diagram in Figure 6 depicts the intersection and difference sets between the Japanese language community and the other language communities. The Japanese difference set includes only hashtags used in the Japanese language community. The intersection set consists of hashtags that appeared in all language communities, and the difference set of the other six languages exhibits hashtags mentioned in all six languages. The hashtags that appeared in all language communities were sorted in descending order of the harmonic mean presented in Table 4.

The highest harmonic means among the intersection hashtags were for #biodiversity and #Biodiversity. The concept of biodiversity appears to be gaining ground. Moreover, the hashtag #nature has been shared worldwide as a common concept with biodiversity. Furthermore, the hashtag #COP15, which denotes interest in biodiversity, has been increasing rapidly around the world in recent years. The hashtag #science is also popular, which may indicate the contribution of science communities such as IPBES to biodiversity.

The difference set of the six languages comprised hashtags generated only in the six languages. The concepts of ecosystem and protection were inseparable from biodiversity conservation and mainstreaming; they also appeared with hashtags related to the names of specific regions, such as Poland (the host country of the UNFCCC-COP19) and Pakistan (the Japanese language community did not use this word).

Figure 7 presents a comparison of common and unique hashtags between the Japanese and the other six language communities. In all comparison sets, #COP10, #Teddy bear, #Pounding, #ngo, and #Events appeared often in the Japanese difference set. Further, the CBD-COP10 had a large impact on the Japanese biodiversity interest. The Japanese government and the Ministry of the Environment in particular played an important role in mainstreaming biodiversity through the adoption of the Aichi Targets and the Nagoya Protocol. The original Japanese hashtag for #Teddy bear was identified as #MISIA by manually checking the original tweets (this may have been an inaccurate translation from Japanese by Google Translate). #MISIA is a well-known singer who served as a CBD-COP10 honorary ambassador [59]. This singer is an influencer committed to the SDGs and biodiversity. This suggests that influencers have a significant potential to motivate people to develop an interest in

biodiversity. #Pounding refers to the Japanese word "kakiage"—a type of tempura consisting of finely chopped and deep-fried seafood and vegetables such as shrimp. Tweets including #Events were checked manually, and it was found that it was also a mistranslation of "ebi-ten", a type of shrimp tempura. This was related to the fact that the shrimp contained in "kakiage" and "ebi-ten" represented a large proportion of imported shrimp, which led to significant concern regarding the destruction of mangrove forests entailed in their harvesting [60]. The hashtag #ngo was seen in many tweets related to event organization and participation, indicating the large role of nonprofit and nongovernmental organizations in biodiversity.

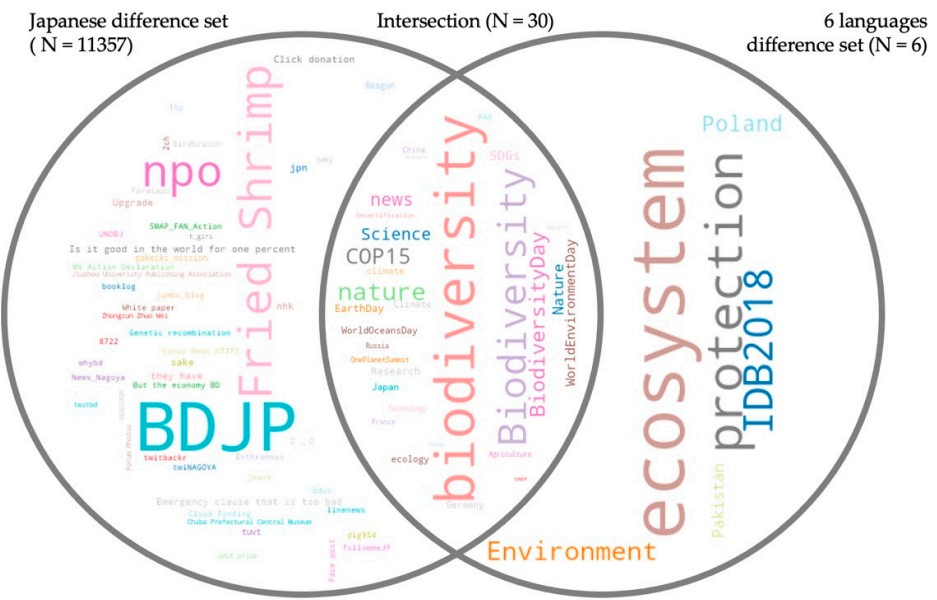

**Figure 6.** Venn diagram of hashtags used among Japanese and the other six languages communities. The Japanese difference set consists of hashtags mentioned only in the Japanese language community; intersection refers to the set of hashtags that appeared in all language communities; and the six languages' difference set includes hashtags that appeared in all six languages. The size of the words displayed in the intersection is proportional to the value of the harmonic mean in Table 5. The number of hashtags displayed in the word cloud was limited to 50 for visibility.

To classify in detail the hashtags observed in the Japanese language community, Table 5 lists the top 10 most prominent hashtags for every five years since 2010. We found a tweet-count explosion in tweets related to biodiversity in 2010 in the Japanese language community. The oldest hashtag trend from 2010 was related to the topics of the CBD-COP10 and specific organizations such as #mudef (a social action group that includes celebrities from various fields, such as artists, actors, and sports players [61]) and #nhk—a state-run broadcasting company (Nippon Hoso Kyokai (NHK)) [62]. Twitter-specific tokens, such as #twvt, were also created as hashtags. The hashtag #twvt stands for a Twitter invite, which is an event and participation announcement service, and #tenen was a hashtag frequently used with #cop10 and #bdjp; however, the exact meaning cannot be clearly identified at this point. The most prominent and autHoritative organizations tended to lead topics in 2010. However, in 2015, contents regarding the relationship between diet and biodiversity (#tempura and #fried shrimp), biodiversity conservation activities by nongovernmental and non-profit organizations (#npo and #ngo), and raising funds for biodiversity conservation (#click donation) were seen more frequently. These issues appear to relate much more closely to citizens' daily lives. As recently as 2020, biodiversity again attracted attention as part of the Global Goals in the SDGs (#SDG). Food systems were also referred to again as lifestyles (#vegan) in 2020.

**Table 4.** Common hashtags in all language communities.

| | Tag | Japanese | Arabic | Chinese | English | French | Russian | Spanish | Overall |
|---|---|---|---|---|---|---|---|---|---|
| | **Intersection (N = 30)** | | | | | | | | |
| 1 | biodiversity | 1.9160 | 0.3686 | 0.9119 | 14.9146 | 16.0186 | 12.6573 | 9.1214 | 1.5055 |
| 2 | Biodiversity | 0.3151 | 10.1794 | 0.1520 | 3.2803 | 2.2924 | 2.6923 | 5.9170 | 0.6287 |
| 3 | nature | 0.1594 | 0.1352 | 0.0507 | 2.2690 | 1.2392 | 0.4196 | 0.6749 | 0.1817 |
| 4 | COP 15 | 0.1314 | 0.4730 | 2.6342 | 0.1358 | 0.1491 | 0.1748 | 0.0454 | 0.1349 |
| 5 | BiodiversityDay | 0.0645 | 0.3686 | 0.1013 | 0.2324 | 0.1309 | 0.4196 | 0.0662 | 0.1217 |
| 6 | news | 0.6204 | 0.1352 | 1.6717 | 0.0966 | 0.0198 | 0.7692 | 0.0811 | 0.0832 |
| 7 | Science | 0.0414 | 0.0430 | 0.0507 | 0.1539 | 0.0590 | 0.1399 | 0.1528 | 0.0671 |
| 8 | Nature | 0.0109 | 0.0676 | 0.0507 | 0.4544 | 0.4106 | 1.3636 | 0.3057 | 0.0520 |
| 9 | WorldEnvironmentDay | 0.0134 | 0.1843 | 0.1013 | 0.2578 | 0.0367 | 0.2448 | 0.0225 | 0.0413 |
| 10 | SDGs | 1.5048 | 0.1290 | 0.0507 | 0.2338 | 0.0147 | 0.1399 | 0.0110 | 0.0353 |
| 11 | Research | 0.0487 | 0.0061 | 0.0507 | 0.0249 | 0.0274 | 0.0350 | 0.0331 | 0.0207 |
| 12 | EarthDay | 0.0049 | 0.0184 | 0.1013 | 0.0928 | 0.0340 | 0.1049 | 0.0299 | 0.0198 |
| 13 | Climate | 0.0049 | 0.0123 | 0.0507 | 0.1384 | 0.3792 | 0.0350 | 0.0178 | 0.0174 |
| 14 | Japan | 0.0255 | 0.0369 | 0.0507 | 0.0151 | 0.0065 | 0.0350 | 0.0138 | 0.0172 |
| 15 | climate | 0.0049 | 0.0061 | 0.0507 | 0.5006 | 0.9797 | 0.5594 | 0.0610 | 0.0171 |
| 16 | Germany | 0.0462 | 0.0491 | 0.0507 | 0.0138 | 0.0053 | 0.1049 | 0.0126 | 0.0170 |
| 17 | WorldOceansDay | 0.0097 | 0.0123 | 0.0507 | 0.0352 | 0.0183 | 0.0699 | 0.0077 | 0.0163 |
| 18 | ecology | 0.0036 | 0.0061 | 0.0507 | 0.3097 | 0.4840 | 2.9021 | 0.3496 | 0.0151 |
| 19 | China | 0.0681 | 0.5099 | 4.5593 | 0.0561 | 0.0018 | 0.0699 | 0.0358 | 0.0112 |
| 20 | FAO | 0.0024 | 0.0184 | 0.0507 | 0.0126 | 0.0125 | 0.6993 | 0.0238 | 0.0102 |
| 21 | Technology | 0.0024 | 0.0061 | 0.1013 | 0.0155 | 0.0042 | 0.0350 | 0.0447 | 0.0075 |
| 22 | Agriculture | 0.0024 | 0.0061 | 0.1013 | 0.0629 | 0.1871 | 0.2098 | 0.0024 | 0.0068 |
| 23 | OnePlanetSummit | 0.0012 | 0.0307 | 0.0507 | 0.0186 | 0.0941 | 0.1748 | 0.0089 | 0.0066 |
| 24 | France | 0.0085 | 0.2273 | 0.1013 | 0.0166 | 0.2568 | 0.0350 | 0.0011 | 0.0063 |
| 25 | Russia | 0.0024 | 0.0246 | 0.0507 | 0.0069 | 0.0044 | 0.3846 | 0.0033 | 0.0061 |
| 26 | Desertification | 0.0036 | 0.0369 | 0.0507 | 0.0045 | 0.0015 | 0.0350 | 0.0088 | 0.0051 |
| 27 | today | 0.0049 | 0.0430 | 0.0507 | 0.0027 | 0.0004 | 0.0699 | 0.0053 | 0.0020 |
| 28 | UNEP | 0.0085 | 0.1167 | 0.0507 | 0.0327 | 0.0004 | 0.1748 | 0.0014 | 0.0019 |
| 29 | COVID19 | 0.0182 | 0.0491 | 0.0507 | 0.1183 | 0.1197 | 0.1748 | 0.0003 | 0.0019 |
| 30 | fornature | 0.0012 | 0.0123 | 0.0507 | 0.0201 | 0.0004 | 0.0699 | 0.0009 | 0.0014 |
| | **Six Languages Difference** | | | | | | | | |
| | **Tag** | **Japanese** | **Arabic** | **Chinese** | **English** | **French** | **Russian** | **Spanish** | **Overall** |
| 1 | Environment | 0 | 0.1904 | 0.0507 | 0.4123 | 0.6253 | 0.0350 | 1.6721 | 0.1031 |
| 2 | ecosystem | 0 | 0.0123 | 0.0507 | 0.2059 | 0.0434 | 0.1049 | 0.0811 | 0.0398 |
| 3 | protection | 0 | 0.0123 | 0.0507 | 0.0158 | 0.0737 | 0.0699 | 0.0147 | 0.0231 |
| 4 | Poland | 0 | 0.0061 | 0.1013 | 0.0029 | 0.0401 | 0.0350 | 0.0034 | 0.0070 |
| 5 | IDB2018 | 0 | 0.0123 | 0.1013 | 0.0041 | 0.0020 | 0.0350 | 0.0029 | 0.0050 |
| 6 | Pakistan | 0 | 0.0061 | 0.1013 | 0.0217 | 0.0013 | 0.0350 | 0.0005 | 0.0020 |

Note: The numbers indicate the proportion of the cumulative number of times the hashtag was mentioned to the total number of hashtags over the sample period for a given language community. The overall value of each row is the harmonic mean of the proportions of all language communities.

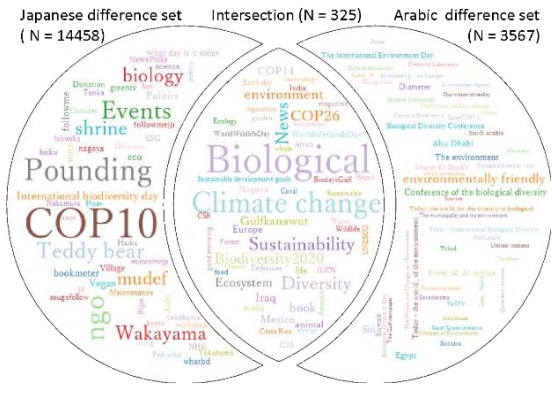

| Japanese difference set | | | |
| --- | --- | --- | --- |
| Tag | Japanese | Arabic | Harmonic mean |
| 1 COP10 | 3.45 | 0 | - |
| 2 Pounding | 1.60 | 0 | - |
| 3 ngo | 1.50 | 0 | - |
| 4 Events | 0.99 | 0 | - |
| 5 Teddy bear | 0.80 | 0 | - |
| Intersection | | | |
| 1 Biological | 37.8 | 2.4 | 4.5 |
| 2 Climate change | 3.1 | 6.3 | 4.2 |
| 3 Sustainability | 1.4 | 1.0 | 1.2 |
| 4 Diversity | 0.8 | 1.8 | 1.1 |
| 5 News | 1.4 | 0.9 | 1.1 |
| Arabic difference set | | | |
| 1 environmentally friendly | 0 | 2.11 | - |
| 2 Conference of the biological diversit | 0 | 2.07 | - |
| 3 Sisi | 0 | 2.05 | - |
| 4 The environment | 0 | 1.66 | - |
| 5 AbuDhabi | 0 | 1.58 | - |

(**a**) Arabic

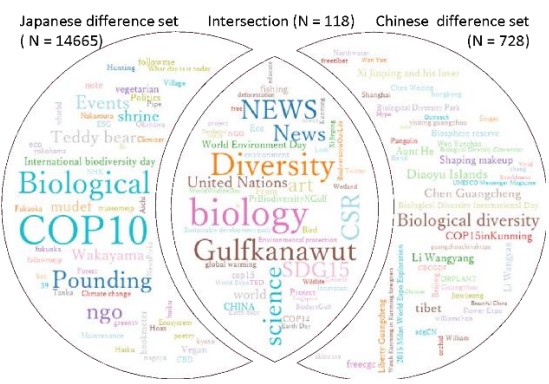

| Japanese difference set | | | |
| --- | --- | --- | --- |
| Tag | Japanese | Chinese | Harmonic mean |
| 1 COP10 | 3.27 | 0 | - |
| 2 Biological | 2.96 | 0 | - |
| 3 Pounding | 1.52 | 0 | - |
| 4 ngo | 1.42 | 0 | - |
| 5 Events | 0.93 | 0 | - |
| Intersection | | | |
| 1 biology | 22.77 | 2.78 | 4.95 |
| 2 Diversity | 2.35 | 4.94 | 3.18 |
| 3 Gulfkanawut | 2.40 | 3.09 | 2.70 |
| 4 NEWS | 1.98 | 2.16 | 2.07 |
| 5 science | 5.17 | 1.23 | 1.99 |
| Chinese difference set | | | |
| 1 Biological diversity | 0 | 3.04 | - |
| 2 tibet | 0 | 1.60 | - |
| 3 Diaoyu Islands | 0 | 1.44 | - |
| 4 2015 Milan World Expo Exploration | 0 | 0.61 | - |
| 5 Beijing | 0 | 0.61 | - |

(**b**) Chinese

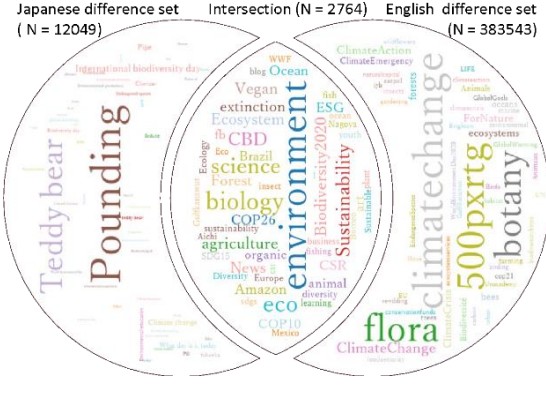

| Japanese difference set | | | |
| --- | --- | --- | --- |
| Tag | Japanese | English | Harmonic mean |
| 1 Pounding | 2.23 | 0 | - |
| 2 Teddy bear | 1.11 | 0 | - |
| 3 International biodiversity day | 0.79 | 0 | - |
| 4 Climate change | 0.36 | 0 | - |
| 5 Pipe | 0.36 | 0 | - |
| Intersection | | | |
| 1 environment | 1.76 | 4.69 | 2.56 |
| 2 biology | 1.74 | 0.53 | 0.81 |
| 3 eco | 1.09 | 0.61 | 0.78 |
| 4 science | 0.40 | 1.55 | 0.63 |
| 5 Sustainability | 0.31 | 0.99 | 0.47 |
| English difference set | | | |
| 1 climatechange | 0 | 1.58 | - |
| 2 500pxrtg | 0 | 1.35 | - |
| 3 flora | 0 | 1.35 | - |
| 4 botany | 0 | 1.21 | - |
| 5 ClimateChange | 0 | 0.97 | - |

(**c**) English

**Figure 7.** *Cont.*

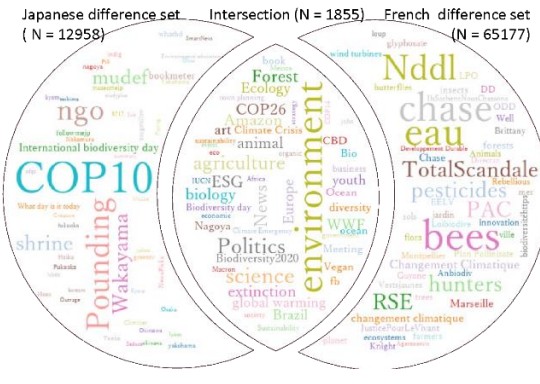

| Japanese difference set | | | |
|---|---|---|---|
| Tag | Japanese | French | Harmonic mean |
| 1 COP10 | 4.09 | 0 | - |
| 2 Pounding | 1.90 | 0 | - |
| 3 ngo | 1.78 | 0 | - |
| 4 shrine | 0.76 | 0 | - |
| 5 Wakayama | 0.75 | 0 | - |
| Intersection | | | |
| 1 environment | 2.71 | 16.62 | 4.66 |
| 2 science | 0.61 | 0.72 | 0.66 |
| 3 Politics | 1.70 | 0.39 | 0.63 |
| 4 agriculture | 0.29 | 3.79 | 0.54 |
| 5 News | 0.50 | 0.34 | 0.40 |
| French difference set | | | |
| 1 bees | 0 | 0.64 | - |
| 2 chase | 0 | 0.61 | - |
| 3 eau | 0 | 0.61 | - |
| 4 Nddl | 0 | 0.45 | - |
| 5 TotalScandale | 0 | 0.44 | - |

(**d**) French

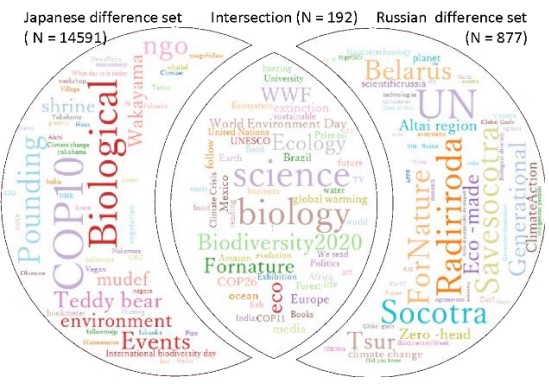

| Japanese difference set | | | |
|---|---|---|---|
| Tag | Japanese | Russian | Harmonic mean |
| 1 COP10 | 3.30 | 0 | - |
| 2 Biological | 2.99 | 0 | - |
| 3 Pounding | 1.53 | 0 | - |
| 4 ngo | 1.43 | 0 | - |
| 5 Events | 0.94 | 0 | - |
| Intersection | | | |
| 1 science | 4.0 | 3.5 | 3.7 |
| 2 biology | 17.6 | 1.9 | 3.4 |
| 3 Biodiversity2020 | 2.5 | 0.9 | 1.4 |
| 4 eco | 11.1 | 0.7 | 1.3 |
| 5 Ecology | 0.9 | 2.3 | 1.3 |
| Russian difference set | | | |
| 1 UN | 0 | 3.22 | - |
| 2 Radiriroda | 0 | 2.29 | - |
| 3 Savesocotra | 0 | 2.07 | - |
| 4 Socotra | 0 | 2.07 | - |
| 5 ForNature | 0 | 1.72 | - |

(**e**) Russian

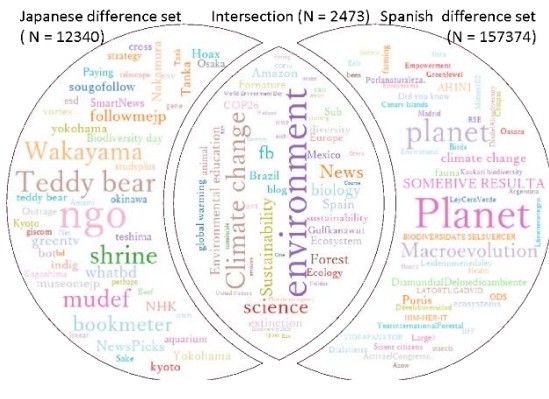

| Japanese difference set | | | |
|---|---|---|---|
| Tag | Japanese | Spanish | Harmonic mean |
| 1 ngo | 1.98 | 0 | - |
| 2 Teddy bear | 1.05 | 0 | - |
| 3 shrine | 0.84 | 0 | - |
| 4 Wakayama | 0.83 | 0 | - |
| 5 mudef | 0.76 | 0 | - |
| Intersection | | | |
| 1 environment | 1.99 | 8.31 | 3.21 |
| 2 Climate change | 0.79 | 3.73 | 1.30 |
| 3 science | 0.44 | 1.83 | 0.72 |
| 4 News | 0.37 | 1.17 | 0.56 |
| 5 fb | 0.44 | 0.68 | 0.54 |
| Spanish difference set | | | |
| 1 Planet | 0 | 1.72 | - |
| 2 planet | 0 | 0.82 | - |
| 3 Macroevolution | 0 | 0.48 | - |
| 4 SOMEBIVE RESULTA | 0 | 0.47 | - |
| 5 climate change | 0 | 0.45 | - |

(**f**) Spanish

**Figure 7.** Comparison of each of the other six languages with Japanese. In all the Venn diagrams, from (**a**–**f**), the Japanese difference set is visualized on the left, the intersection set between Japanese and the compared language community is in the center, and the compared language community's difference set is on the right. The 30 most common hashtags in Table 4 were excluded from the intersection set. The number of hashtags displayed in the word cloud was limited to 50 to ensure readability.

**Table 5.** Transition of popular hashtags in the Japanese language community.

| Rank | 2010 | 2015 | 2020 |
|---|---|---|---|
| 1 | bdjp | Tempra | biodiversity |
| 2 | cop10 | Fried shrimp | SDGs |
| 3 | biodiversity | npo | note |
| 4 | COP10 | ngo | news |
| 5 | MISIA | biodiversity | International biodiversity day |
| 6 | tenen | Click donation | vegan |
| 7 | mudef | bookmeter | vegan |
| 8 | gakeiki_mission | news | living things |
| 9 | twvt | Kyushu University press | vegetarian |
| 10 | nhk | environmental protection | environment |

Note: Japanese hashtags were manually translated into English by the author.

### 3.3.3. Correspondence with IPBES-CF

The hashtags in the intersection set of the seven language communities presented in Table 4 were grouped with reference to the IPBES-CF category, as presented in Table 6. The indirect drivers covered 53% of the total hashtags. These drivers could be further categorized into international events and initiatives (#Biodiversityday, #WorldEnvironmentDay, #EarthDay, #WorldOceanDay, and #fornature), institutions and governance (#SDGs, #COP15, and #OnePlanetSummit), countries and organizations (#Japan, #Germany, #China, #FAO, #France, #Russia, and #UNEP), and current events (#news). Nature was found in 23% of the hashtags. The two categories of nature commonly observed were climate (#Climate and #climate) and ecosystems (#biodiversity, #Biodiversity, #nature, #Nature, and #ecology). The anthropogenic assets and direct drivers' categories followed with 10% each. The rest referred to NCP and the good quality of life associated with NCP. Neither category had any hashtag within the intersection hashtag set. The hashtag #today, categorized among other views, was used to indicate biodiversity-related events and happenings at that time.

**Table 6.** Comparison of hashtags and IPBES-CF.

| Framework | Proportion of Tags (N = 31) | Tag Examples |
|---|---|---|
| 1 Nature | 0.23 | biodiversity, Biodiversity, nature, Nature, Climate, climate, ecology |
| 2 Nature's benefits to people | 0.00 | - |
| 3 Anthropogenic assets | 0.10 | Science, Research, Technology |
| 4 Indirect drivers | 0.53 | COP 15, BiodiversityDay, news, WorldEnvironmentDay, SDGs, EarthDay, Japan, Germany, WorldOceansDay, China, FAO, OnePlanetSummit, France, Russia, UNEP, fornature |
| 5 Direct drivers | 0.10 | Agriculture, Desertification, COVID19 |
| 6 Good quality of life | 0.00 | - |
| Other views | 0.03 | today |

Note: the hashtags listed in "Tag examples" are given in descending order of the overall values presented in Table 4.

Table 7 presents the proportion of the IPBES-CF categories for the 117 hashtags in the Japanese and English language communities, corresponding to the highest 1% of hashtags used only in the Japanese language community and the highest 0.03% of hashtags used only in the English language community. In all, 67% of hashtags in the Japanese language community were grouped under the indirect drivers' category. The common hashtags included frequently used abbreviations in Japan for the names

CBD-COP10 (#BDJP), national broadcasting company (#nhk), #click donations [63], and the youth special mission to realize the Aichi Targets (#gakeiki_mission) [64].

**Table 7.** Comparison of unique hashtags of Japanese and English language communities according to the conceptual framework element.

| Framework | Proportion of Tags N = 117 | | |
|---|---|---|---|
| | **Japanese** | **English** | ***p* Value** |
| 1. Nature | 0.03 | 0.09 | 0.098 |
| 2. Nature's contributions to people | 0.02 | 0.03 | 0.408 |
| 3. Anthropogenic assets | 0.07 | 0.11 | 0.253 |
| 4.I ndirect drivers | 0.67 | 0.56 | 0.107 |
| 5. Direct drivers | 0.04 | 0.16 | 0.003 |
| 6. Good quality of life | 0.03 | 0.04 | 0.472 |
| Other views | 0.15 | 0.00 | 0.000 |

A $\chi^2$ test detected a significant difference in the probability distribution between the language communities at the 5% level, and a residual test exhibited a difference in meaning in the probability of other views ($p = 0.000$) and direct drivers ($p = 0.003$). In the Japanese language community, 15% of the hashtags cited biodiversity from a different perspective other than IPBES-CF. Regarding the direct drivers, the English language community included all direct drivers defined by a transformative change study—"Land/sea-use change", "Direct exploitation", "Climate change", "Pollution", "Invasive species", and "Conservation activities" [65]. By contrast, the Japanese language community included hashtags in limited categories such as invasive species (#invasive species), direct exploitation (#illegal deal), and bird conservation (#Bird-Branch). The IPBES-CF was developed by international experts in an open process and widely shared, but this result suggests that there is still room for improvement.

### 3.3.4. Network Metrics

Table 8 presents the hashtags with the highest betweenness centrality among the hashtags of the communities with the highest number of hashtags, the hashtags translated into English by Google Translate, and the betweenness centrality values. More than three proper nouns, such as organization names and international conferences, were extracted from the Japanese, Arabic, Chinese, and Russian language communities. In addition, more than four nouns paraphrasing biodiversity were found as hashtags with high betweenness centrality in the English, French, Russian, and Spanish language communities. The Japanese language community's uniqueness was that hashtags related to media such as the state-run broadcasting company and news had high betweenness centralities.

Table 9 presents the time series of the highest betweenness centrality hashtags in the Japanese language community. In terms of hashtag content, until 2011, hashtags regarding the CBD-COP10 were the focus. From 2012 to 2017, biodiversity (or paraphrasing of biodiversity) became the focus of interest. Education-related topics, such as #Kindle and #environmental education, had higher betweenness centralities. After 2018, sustainability topics other than biodiversity, such as #SDGs, #ESG, and #Zerohunger, were also connected to biodiversity and various topics.

**Table 8.** The highest betweenness centers in each language community.

|   | Japanese | Betweenness Centrality |   | Arabic | Betweenness Centrality |   | Chinese | Betweenness Centrality |   | English | Betweenness Centrality |
|---|---|---|---|---|---|---|---|---|---|---|---|
| 1 | biodiversity | 0.37 | 1 | Diameter | 0.14 | 1 | COP15 | 0.37 | 1 | Biodiversity | 0.15 |
| 2 | cop10 | 0.25 | 2 | Kuwait | 0.07 | 2 | biodiversity | 0.14 | 2 | Wildlife | 0.06 |
| 3 | Nagoya | 0.13 | 3 | Abu Dhabi | 0.07 | 3 | Biological diversity | 0.13 | 3 | Nature | 0.05 |
| 4 | surroundings | 0.13 | 4 | UAE | 0.06 | 4 | COP15 in Kunming | 0.09 | 4 | nature | 0.04 |
| 5 | Quiet life | 0.12 | 5 | Egypt | 0.06 | 5 | Conservation | 0.07 | 5 | Environment | 0.04 |
| 6 | BDJP | 0.11 | 6 | Climate change | 0.05 | 6 | SDG15 | 0.05 | 6 | Conservation | 0.02 |
| 7 | news | 0.09 | 7 | Today-the world_of the environment | 0.05 | 7 | Connect2Earth | 0.02 | 7 | climate change | 0.02 |
| 8 | COP10 | 0.08 | 8 | Sisi | 0.05 | 8 | Endangered Species | 0.02 | 8 | conservation | 0.01 |
| 9 | biology | 0.07 | 9 | France | 0.04 | 9 | Hype | 0.02 | 9 | environment | 0.01 |
| 10 | nhk | 0.07 | 10 | Sharm El-Shaikh | 0.03 | 10 | BRI | 0.02 | 10 | wildlife | 0.01 |

|   | French | Betweenness Centrality |   | Russian | Betweenness Centrality |   | Spanish | Betweenness Centrality |
|---|---|---|---|---|---|---|---|---|
| 1 | Nature | 0.07 | 1 | ecology | 0.13 | 1 | Biodiversity | 0.48 |
| 2 | Biodiversity | 0.07 | 2 | Biodiversity | 0.12 | 2 | Environment | 0.05 |
| 3 | biodiversity | 0.05 | 3 | FAO | 0.05 | 3 | Nature | 0.05 |
| 4 | Environment | 0.03 | 4 | biodiversity | 0.04 | 4 | nature | 0.03 |
| 5 | Ecology | 0.02 | 5 | Tsur | 0.04 | 5 | app | 0.02 |
| 6 | environment | 0.02 | 6 | AND | 0.04 | 6 | Corlantatura | 0.02 |
| 7 | Hunt | 0.01 | 7 | soil | 0.03 | 7 | Space | 0.02 |
| 8 | PAC | 0.01 | 8 | Nature | 0.03 | 8 | Wetlands | 0.02 |
| 9 | Marseille | 0.01 | 9 | UN | 0.03 | 9 | environment | 0.02 |
| 10 | animals | 0.01 | 10 | scientificrussia | 0.03 | 10 | conservación | 0.01 |

**Table 9.** Time series of hashtags with the highest betweenness centralities in the Japanese language community.

|   | Hashtags | Between Centrality | Hashtags | Between Centrality | Hashtags | Between Centrality |
|---|---|---|---|---|---|---|
|   | 2009 | | 2010 | | 2011 | |
| 1 | bdjp | 1.000 | cop10 | 0.398 | cop10 | 0.436 |
| 2 | | | bdjp | 0.291 | bdjp | 0.308 |
| 3 | | | biodiversity | 0.171 | kaminoseki | 0.245 |
| 4 | | | COP10 | 0.160 | biodiversity(inEnglish) | 0.234 |
| 5 | | | tokyo | 0.089 | biodiversity(chinesecharacter) | 0.193 |
| 6 | | | eco | 0.079 | genpatsu | 0.145 |
| 7 | | | CBD | 0.073 | COP10 | 0.098 |
| 8 | | | nagoya | 0.068 | CBD | 0.085 |
| 9 | | | followdaibosyu | 0.053 | kankyo | 0.073 |
| 10 | | | followmejp | 0.052 | 1 year passed after COP10 | 0.056 |

**Table 9.** *Cont.*

| | Hashtags | Between Centrality | Hashtags | Between Centrality | Hashtags | Between Centrality |
|---|---|---|---|---|---|---|
| | 2012 | | 2013 | | 2014 | |
| 1 | biodiversity | 0.659 | nature | 0.446 | biodiversity | 0.349 |
| 2 | environment | 0.483 | living things | 0.382 | npo | 0.282 |
| 3 | yokohama | 0.138 | environment | 0.275 | environment | 0.266 |
| 4 | clam | 0.065 | biodiversity | 0.218 | ngo | 0.256 |
| 5 | yokohama | 0.065 | ngo | 0.217 | sea | 0.101 |
| 6 | kankyo | 0.065 | npo | 0.217 | livingthing | 0.077 |
| 7 | eco | 0.065 | CSR | 0.043 | ESD | 0.027 |
| 8 | living things | 0.039 | Henoko | 0.043 | science | 0.026 |
| 9 | science | 0.039 | event | 0.043 | COP12 | 0.026 |
| 10 | animal | 0.039 | science | 0.016 | kindle | 0.026 |
| | 2017 | | 2018 | | 2019 | |
| 1 | biology | 0.527 | biodiversity | 0.616 | biodiversity | 0.871 |
| 2 | environment | 0.495 | biology | 0.388 | My world through a viewfinder | 0.143 |
| 3 | nature | 0.495 | International day for biodiversity | 0.314 | extinction | 0.132 |
| 4 | blog | 0.143 | Okinawa | 0.256 | crowdfunding | 0.121 |
| 5 | | | SDGs | 0.221 | palm oil | 0.120 |
| 6 | | | environment | 0.129 | International day for biodiversity | 0.104 |
| 7 | | | nature | 0.128 | Borneo | 0.094 |
| 8 | | | Henoko | 0.124 | Climate Change | 0.070 |
| 9 | | | ESG | 0.062 | SDGs | 0.064 |
| 10 | | | What day is today | 0.032 | ZeroHunger | 0.059 |
| | 2020 | | 2021 | | | |
| 1 | biodiversity | 0.862 | biodiversity | 0.636 | | |
| 2 | International day for biodiversity | 0.114 | news | 0.238 | | |
| 3 | animal | 0.093 | SDGs | 0.165 | | |
| 4 | climate crisis | 0.085 | living things | 0.128 | | |
| 5 | living thing | 0.082 | environment | 0.106 | | |
| 6 | news | 0.071 | diversity | 0.075 | | |
| 7 | Climate Crisis | 0.058 | International day for biodiversity | 0.060 | | |
| 8 | Naniwa eco meeting | 0.052 | climate change | 0.058 | | |
| 9 | Hanno | 0.052 | animal | 0.058 | | |
| 10 | Aichi | 0.051 | satoyama | 0.045 | | |

## 4. Discussion

### 4.1. RQ1: Is the Interest in Biodiversity Continuously Activated (RQ1-1) and Diversified (RQ1-2)

In this study, we found that discussions on biodiversity are becoming more active in all seven language communities, as indicated by the trend of an increasing number of tweets (Figure 4) and engagements (Figure 5a–c), providing an affirmative answer to RQ1-1. In the Japanese language community, the number of tweets (Figure 4) has been increasing since 2015, and engagement (Figure 5a–c) was increasing until around 2019. Subsequently, engagement of the Japanese language community slowed after 2019, but the number of tweets increased, suggesting that Twitter became an active forum for biodiversity discussions.

The answer to RQ1-2 is also yes. The rate of emergence of unique hashtags (Table 2) increased, and the diversity of the interest increased almost monotonically (Figure 5d). This indicates that the global biodiversity conversation is becoming more active and diverse; views on the value of biodiversity are becoming more diverse; and stakeholders' activities in biodiversity conservation are present in various fields. This results in a different understanding of the role that nature plays at the foundation of people's lives and in their quality of life, leading to the formation of a wide diversity of values regarding nature [2].

From an international perspective, the Japanese language community has been relatively inactive in discussions that deal directly with biodiversity topics. Japan has the second-largest number of Twitter users in the world [66]; however, the number of tweets related to biodiversity was fourth among the seven language communities. This can be interpreted in two ways: the Japanese language community had a weak interest in biodiversity, or the Japanese language community does not use the word "biodiversity", using other words to express this concept, such as "natural richness" or "gift from nature". We will discuss this in the section below regarding limitations.

### 4.2. RQ2: What Are the Shared Interests among the Language Communities and the Special Interest of the Japanese Language Community?

Along with the IPBES-CF, all language communities expressed great interest in the indirect drivers' category (Table 6). International days such as World Environment Day, Biodiversity Day, Earth Day, and World Oceans Day provided notable biodiversity content and attract attention. Zarrabeitia et al. (2022) found that these international days play a large role in positive discussions with collective sentiment [67]. Nature was the second most popular category observed with high betweenness centrality hashtags. This suggests that recapturing and disseminating nature value is effective in stimulating biodiversity discussions on Twitter. Twitter users are also interested in the relationship between the climate and ecosystems, the importance of which was strongly emphasized by IPBES and IPCC [68]. Biodiversity issues are expected to be simultaneously resolved through mitigation of and adaptation to climate change. Among anthropogenic assets, the science and technology to conserve biodiversity is universally recognized as the key in all language communities, indicating that IPBES has a significant role to play in biodiversity mainstreaming. In the direct drivers' category, the interlinkage between food, biodiversity, and the impacts of desertification, land modification, and COVID-19 were addressed. In particular, human destruction of ecosystems and unsustainable use of NCP enlarge the risk of pandemics, as represented by the COVID-19 pandemic [69–71]. Hashtags such as #COVID19 and #OnePlanetSummit suggest that Twitter users focus on biodiversity conservation, which has synergistic effects on the prevention of infectious diseases.

As a feature of the uniqueness of the Japanese language community, it expressed little concern regarding the direct drivers. This result implies that the Japanese language community may have a weak perspective not only on individual species but also on conserving the ecosystem as a whole [72]. In particular, in the other-views category, the Japanese language community used hashtags that focused on biodiversity

from a different perspective. For example, #Kaminoseki, the proposed construction site for a nuclear power plant, is a valuable habitat for the *Branchiostoma japonicum* and the black heron (*Gerres equulus*) [73], and opponents of nuclear power referred to biodiversity issues from the perspective of energy policy. This type of interlinkage is also important from the perspective of the biodiversity–energy nexus issues, and it will be a key issue in renewable energy mainstreaming [74]. Another hashtag, #Life After, mentioned a game with a system in which biodiversity conservation in the game process provides many benefits to its players. This suggests an interesting potential that the metaverse, including virtual reality games, can play an effective role in mainstreaming biodiversity [75,76].

### 4.3. RQ3: What Promotes the Interest in Biodiversity in the Japanese Language Community?

In 2010 and 2020, the main concern of the Japanese language community shifted from the hashtags of global top-down elements to local bottom-up elements (Table 5). In the early era of 2010, as individual organizations (#mudef and #nhk) and celebrities (#MISIA) appeared frequently, it indicated that media and influencers provided a considerable contribution to promoting the public conversation on biodiversity. The tweets revealed the most activity during the entire sample period (Figures 4 and 5). This phenomenon, in which certain features rose sharply in frequency as the topic emerged, is called a "burst" of activity [77], which was triggered by major events such as earthquakes, elections, or fires [78]. On the other hand, the diversity index was not high around 2010 (Figure 5d), and hashtags related to the CBD-COP10 accounted for 32.6% of hashtags. In addition, the hashtags #bot and #recobot were used frequently. A "recobot" is an account that analyzes the words of a user's reply to it to understand the user's needs and searches for recommended products and information [79]. This type of bot may have been associated with this burst of tweets [80] and can have an impact on the formation of public opinion [81,82].

From 2015 to the present, hashtags for citizen-driven organizations, such as #npo and #ngo, have been used. These agents are innovators in increasing public awareness of biodiversity. From 2015 (oysters and fried shrimp) to 2020 (vegans), commitment to examining the relationship between food systems and biodiversity remained high. Growing biodiversity issues regarding the concept of the planetary health diet can stimulate discussions more broadly [83]. Furthermore, in 2020, the most recent year in the data, #SDG became one of the most used hashtags. SDGs may be growing as a hub of sustainability issues in the biodiversity domain. Viewing biodiversity not as a stand-alone environmental issue but as an issue connected with other environmental and socioeconomic issues inclusively is key to mainstreaming it.

### 4.4. Limitations

This study has five main limitations. The first relates to bias and accuracy problems that arose from the use of machine translation with Google Translate. Translations using the Google Translate API sometimes do not correctly capture the meaning of the original language. Reports of such errors are updated on the Google Help page from time to time [84]. For example, the Japanese hashtag "#kankyo", which means "environment" written in the Roman alphabet, was translated to English as "pipe". The hashtags also include many slang terms or informal language that is only widespread on Twitter, making the translation more complex. Previous studies have used machine translation with Google Translate or analyzed only English hashtags, but this is an important concern in global Twitter analysis that includes various local languages [85]. Another issue is determining appropriate preprocessing and morphological analysis, such as the lower method, substitution, and splitting of connective words at the point of translation into English. This kind of preprocessing can be a bias factor, so we can apply good techniques of using a multi-language tokenizer and lemmatizer coupled with multilingual emotion analysis as shown in Basile (2021) [86].

Second, interpreting the content was also an issue. Our research team understood original tweets in Japanese and English only. Therefore, tweets in other languages could not be investigated in detail because the authors were only able to interpret the texts of the tweets through machine translation. Various problems with the interpretation of social media data have been raised previously [87,88]. These obstacles are expected to be alleviated by having native speakers of each language participate and discuss the tweets together.

Third, as it was only possible to access publicly available tweets at the time of the search, the dataset changed at each search [89,90]. Social networks are dynamic by nature. Users add, change, and delete their content at any time. If a user deletes individual tweets or their entire account, tweets about biodiversity that they posted will also become inaccessible. Thus, quite unlike physical media, digital media are highly variable and fragile, which means that the results will change depending on when the data were saved. We may be able to avoid investigation bias by focusing on a shorter period and frequent collection is required. Notably, Twitter ended the services in July 2023, so the dataset may be dramatically changed before and after the change. This may be a great bias potential for the future studies.

Fourth, only tweets filtered by the query could be extracted. Tweets that did not include the query term "biodiversity" but included tokens that were equivalent to the concept of biodiversity could not be retrieved by the search. A search that uses concepts that are ontologically similar at the semantic level or machine-learning-based natural language processing technology could compensate for this [91,92]. For example, in Japanese, words such as "nature", "creatures", and "ecosystem" have comparable definitions.

Fifth, the topic mining and the co-occurrence of interest were not discussed in this study. In this study, we analyzed and evaluated the plain hashtags themselves for identification of the biodiversity interest. However, the detection of the topics that consisted of several hashtags would be very informative to the interest's detection. The potential solutions for these issues are both topic-modeling technology, which was applied for biodiversity interests in Ohtani (2022) [44], and artificial intelligence-based pattern matching and community detection technology, which were applied in Cauteruccio et al. (2022) [93], Cauteruccio, F. and Terracina, G. (2023) [94], and Taecharungroj (2023) [95]. In addition, co-occurrence analysis is promising. Analyzing topics likely to be discussed at the same time can help identify new interests with strong connections to biodiversity, such as those related to food, water, health, etc. Such an approach is expected to support the analysis of biodiversity nexuses [96].

Overall, the novelty of this research is that our study conducted a quantitative analysis that did not use people's stated preference data derived from well-structured surveys with limited participants, such as in [11], but used revealed preference data partially proxy to their interests and quantified the historical trend of biodiversity mentions. These results can support comparative analyses with other related surveys conducted in many language communities. This kind of research to monitor the interests using big data on the web is still limited at this point; however, it is a very informative way to evaluate the contributions of various interventions and countermeasures to promote mainstreaming biodiversity in societies.

## 5. Conclusions

This study investigated the commonalities among seven language communities (six official languages of the United Nations and the Japanese language) and the uniqueness of the Japanese language community with respect to biodiversity by comparing tweets that included the term biodiversity and clarifying changes in the interest and concern for biodiversity from the past to the present. The number of tweets and hashtag statics in each language community were compared for tweets that included "biodiversity" in the six official UN languages and the Japanese lan-

guage. This study evaluated the level of engagement and diversity of each language community, and it identified perspectives on biodiversity common among the seven language communities or unique to each language community. In particular, the Japanese language community was found to view the relationship between biodiversity and humans from a different perspective than that presented in the scope of the CF. The approach of this study can be useful to identify and monitor the progress of mainstreaming biodiversity and facilitate the communication of shared interests and uniqueness, depending on the local context. It can also present effective interventions and drivers for enhancing interest in biodiversity. In this study, we focused on the uniqueness of the Japanese language community, but we can easily apply the same procedure to other languages to discover their own views on biodiversity.

The SDGs are a set of commonly agreed-upon global goals that establish targets for marine and terrestrial biodiversity in Global Goals 14 and 15, respectively. The SDGs require comprehensive solutions to all international goals in a co-operative manner. The Kunming-Montreal Global Biodiversity Framework, which follows the Aichi Targets, also proposes the importance of the inclusion of biodiversity values in all sectors [6,7]. It is necessary to group tweets by the co-occurrence of interest to clarify the inter between a passion for biodiversity and the SDGs in future work. In addition, although this analysis focused on the Japanese language community, collaborating with scholars from communities around the world is desirable for understanding the concept of biodiversity across traditions and cultures.

**Author Contributions:** S.I. and C.H. carried out the collection of the tweet database; S.I., C.H. and T.M. worked out almost all the technical details, developed the theoretical formalism, performed the analytic calculations, and performed the numerical simulations; K.H., S.H. and O.S. provided the latest knowledge and deepened the discussion; S.I. and T.M. conceived the original idea and created the draft manuscript; all authors joined the conceptualization and contributed to the finalization. All authors have read and agreed to the published version of the manuscript.

**Funding:** This work was supported by the Environment Research and Technology Development Fund (S-21, JPMEERF23S12100, and 1-2104, JPMEERF20211004) of the Environmental Restoration and Conservation Agency of Japan and by the e-ASIA JRP (Grant Number JPMJSC20E6) of the Japan Science and Technology Agency. This manuscript has not been published elsewhere and is not under consideration by another journal.

**Institutional Review Board Statement:** Not applicable.

**Informed Consent Statement:** Not applicable.

**Data Availability Statement:** Data sharing not applicable.

**Conflicts of Interest:** The authors declare no conflict of interest.

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
