# Peer review of "Twitter Mining for Detecting Interest Trends on Biodiversity: Messages from Seven Language Communities"

_sustainability, doi:10.3390/su151712893_

Round 1
Reviewer 1 Report (Previous Reviewer 3)
Social media has emerged as a novel information source and a contemporary tool for monitoring public sentiment concerning human-nature interactions. This paper specifically gathers Tweets from seven distinct language communities on the Twitter platform, with a primary focus on biodiversity. By mining and analyzing these collected Tweets, the paper aims to uncover and describe the trends of interest regarding biodiversity.
However, the timeframe for data collection in the study, spanning from 2006 to 2021, raises concerns about the validity and reliability of the data. Many of the contents, links, and other data sources may have been deleted or changed over the years, which could compromise the accuracy of the study's findings.
Social networks are dynamic by nature. Users add, alter, delete content. Thus, many of the older tweets were deleted, and the language used was changed.
The authors acknowledge in section 2.1 that the dataset lacks tweets that were previously deleted and are no longer accessible for collection. Consequently, this omission impairs the validity of the study results, making it impossible to draw results and conclusions.
Therefore, based on the aforementioned concerns, my decision to reject the manuscript.
Author Response
Thanks for your informative suggestion. We agree on the characteristics of fresh and fragile, and also we think it is important to archive a snapshot of the biodiversity interests history.
At the same time, As we know, the policy of Twitter API services has changed largely during this submission and now it requires big cost. As some social network researchers declared to retire the Twitter research, this may be a big loss and tragedy for academia.
Reviewer 2 Report (Previous Reviewer 4)
The paper entitled " Twitter Mining for Detecting Interest Trends on Biodiversity: Messages from Seven Language Communities” studies proposed commonalities among seven language communities. I found the article to be interesting and insightful and particularly well-revised.
Specific Comments:
Abstract: No comments, well written.
Introduction
1. provide useful information for the readers. It is nice to highlight the motivation of the study.
2. The author mentioned the research questions. It is a nice way, however, it is suggested to add the contribution. This will help the non-experts to caught the contribution part easily.
Materials and Methods
1. The author must fill the empty spaceon page 4, and 12. It is suggested that the author move the sections and paragraph to page 4.
2. The author must keep the same flow for captions. I,e. Fig. 2 or Figure 2?
3. How the author crawl the data? What are the expertise of the data collector and annotator? when talking about the annotator experts, it would be much better if authors add some details regarding where the experts come from and what types of experiences they have.
4. The details about the annotation are missing. For instance, what is the inter-rater agreement score? How many people annotated how many data points?
5. It is nice if the author provides some instance examples.
Results
1. I recommend revising the results section, as it is not appropriately managed. However, the authors make many statistical analysis.
2. The author must explain what method is used for Twitter Mining (I,e Deep learning), if done manually, what are the procedure? and finally, the collected data's evaluation score. I recommend to use the F1-score, Precision, Recall, and Accuracy.
Conclusions
No comments, well written
The subject is relevant and will be interesting to the people who read the journal. The author wrote the manuscript very well. Thus, I recommend accepting the paper after major revision.
Author Response
We would like to thank the four reviewers for the constructive and insightful comments. We have carefully examined them, and have made modifications to the manuscript where necessary. All modified texts in the manuscript are tracked with the function of MS-Word and major modification parts are highlighted in red character and yellow and blue highlight. The table below shows our responses to the reviewer’s comments as well as the locations where modifications to the manuscript were made accordingly. The certification of the proofreading is below.
Reviewer 3 Report (Previous Reviewer 2)
What are the direct drivers of change in nature with the greatest global impact according to the proposed framework for transformative change by the Intergovernmental Science-Policy Platform on Biodiversity and Ecosystem Services (IPBES)?
What are the underlying factors called indirect drivers that support the direct drivers of change in nature, as mentioned in the framework for transformative change?
How can transformative change be achieved from the current trend to a more sustainable one, considering both indirect and direct factors, values, behaviors, and governance?
What are the eight leverage points and five levers for transformative change in global sustainability pathways, as illustrated in Figure 1?
How can social media, particularly Twitter, be utilized as a tool for monitoring public opinion and interest in biodiversity, and what insights can be gained from analyzing Twitter data in relation to biodiversity?
How was the dataset for the second collection in February 2022 built, and what limitations should be considered regarding sampling bias?
What is the purpose of hashtag extraction in tweets related to biodiversity, and how were hashtags extracted from the original tweets?
How were engagements and diversity indices evaluated for the hashtags in the dataset, and what research questions did they address?
What is the concept of Shannon entropy, and how was it used as a diversity index to analyze the interest in biodiversity?
How were the most common hashtags visualized and compared among different language communities, and what research questions did this analysis address?
How were the popular hashtags compared with the IPBES Conceptual Framework, and what insights were gained from this comparison?
How did the usage of hashtags differ between the Japanese and English language communities in terms of frequency and variety?
What percentage of hashtags used in tweets about biodiversity were accounted for by the top 10% most used hashtags in the English, French, and Spanish language communities?
How did the engagement levels (RT, like, reply) and diversity index vary across different language communities over time, specifically in relation to topics related to biodiversity?
The quality of English language usage appears to be generally good, with coherent sentence structures and appropriate vocabulary choices. However, there are a few instances where the wording could be improved for better clarity or precision. Overall, while the quality of English language usage in the provided text is generally good, there are a few areas where clarification or refinement could enhance the precision and clarity of the statements.
Author Response
We would like to thank the four reviewers for the constructive and insightful comments. We have carefully examined them, and have made modifications to the manuscript where necessary. All modified texts in the manuscript are tracked with the function of MS-Word and major modification parts are highlighted in red character and yellow and blue highlight. The table below shows our responses to the reviewer’s comments as well as the locations where modifications to the manuscript were made accordingly. The certification of the proofreading is below.
Reviewer 4 Report (New Reviewer)
The following are my comments for improvement of this paper:
1. Figures 1 and 3 are figures which were not generated by the authors. Please state if these figures are available as per the CC BY 4.0 license or similar for direct reuse
2. The authors state – “Twitter API v2 was used to collect tweets…..” in Section 2.1. Please re-write this section to clearly discuss the step-by-step methodology that was followed for data collection. Specifically, the authors should state whether they used the Standard Search API or the Advanced Search API of Twitter. As the Twitter API has rate limits, this section should also present the step(s) which were taken to address the rate limits
3. On page 3, the authors discussed different applications of social media mining and analysis, such as smart cities, traffic, and energy. In the last few years, sentiment analysis and topic modeling have been popular areas of work in this field. Consider briefly discussing sentiment analysis and topic modeling in this context with supporting references such as https://doi.org/10.3390/bdcc7020116 (suggested citation for sentiment analysis) and https://doi.org/10.3390/bdcc7010035 (suggested citation for topic modeling)
4. The specific keywords, phrases, and/or hashtags that were used for data collection from Twitter should be clearly stated.
5. In the context of working with Tweets, how was bot-generated content detected and eliminated? For instance, if a bot account on Twitter tweets something very negative about biodiversity several times – this could affect the overall results of sentiment analysis as well as other aspects of the data analysis. How were such scenarios addressed?
6. The limitations of the study should be clearly stated
Author Response
We thank the new two reviewers for their constructive and insightful comments. We tried our best to reply to the comments. We hope our response fits your suggestions.
Reviewer 5 Report (New Reviewer)
This paper focuses on global change in nature, which is unprecedented. Social media, like Twitter, helps monitor public views on human-nature interactions. This paper proposes a study on biodiversity-related tweets from seven language communities highlights unique perspectives.
The paper focuses on an interesting and important context. The methodology is clear enough and the collected data seems solid. The discussion is thorough and offers different perspectives. However, in the current state the paper surely needs different substantial improvements. In the following, some issues that should be addressed are reported, along with corresponding suggestions:
- Research Questions should be expanded and detailed in a more comprehensive way.
- The pseudocode of Fig. 2 could be adjusted in a different and single Algorithm/Figure.
- The authors correctly highlight that sampling bias could occur. Nevertheless, it seems that the authors do not address this point later. A more detailed discussion should be provided.
- There are several typos in the presentation of variables and formulas; for instance, in 2.3.1, the subscript i is detached from the related formula in the text. The authors should fix all of these typos.
- The authors report that “The translations to English were accepted without evaluating the accuracy of the translation due to limitations of linguistics and cultural understanding”. However, this is not sufficient in general. The authors should provide references to papers carrying out similar tasks with translations (for instance, see [c])
- Table 3 reports some “weird” hashtags (e.g., “Conference of the biological di”). What are these? Are they correct?
- As for related works, there are different works on pattern mining that could be actually interesting to adapt to this context [a,b] The authors could consider citing them.
References:
[a] "Extraction and analysis of text patterns from NSFW adult content in Reddit." Data & Knowledge Engineering 138 (2022): 101979.
[b] "Extended High-Utility Pattern Mining: An Answer Set Programming-Based Framework and Applications." Theory and Practice of Logic Programming (2023): 1-31.
[c] "How dramatic events can affect emotionality in social posting: The impact of COVID-19 on Reddit." Future Internet 13.2 (2021): 29.
Few typos and grammar errors are present.
Author Response
We thank the new two reviewers for their constructive and insightful comments. We tried our best to reply to the comments. We hope our response fits your suggestions.
Round 2
Reviewer 1 Report (Previous Reviewer 3)
Thanks for your feedback. I recognize the dynamic nature of social media, where users add, change, and delete content. As a result, it is inevitable that older tweets are not available and have deleted them. This poses a major challenge in capturing a comprehensive snapshot of the history of biodiversity needed to understand and conserve natural ecosystems.
The non-existence of tweets that are no longer accessible for collection creates a gap in the dataset. Consequently, this omission undermines the validity of the study results, as it makes it impossible to draw precise conclusions and obtain meaningful information.
Considering the above concerns, I have carefully evaluated the manuscript and unfortunately must stand by my decision to reject it. The limitation imposed by the unavailability of the data means that they are not accurate and reliable, which makes it difficult to conduct a detailed analysis that meets the required standards.
I appreciate your effort in contributing to this area of research and recommend that you explore alternative approaches or datasets that may provide more robust and comprehensive results in future studies.
Thanks again for the submission.
Author Response
As your suggestion, web data is quite vulnerable and fragile, so it isn't easy to ensure perfect reproducibility sometimes. Your concern is quite reasonable but we think that our intention to archive the historical snapshots by transparently showing all analytical processes is also important for "Stand on the shoulders of giants". We would like to say thank you for your long commitment to our submission.
Reviewer 2 Report (Previous Reviewer 4)
The authors revised the revised manuscript title "Twitter Mining for Detecting Interest Trends on Biodiversity: 2 Messages from Seven Language Communities" in proper way.
Specific Comments:
Abstract: No comments, well written.
Introduction
1. No comments, well explain and well written.
Realated Works
1. No comments, well explain and well written.
Methods
1. No comments, well explain and well written.
Case of study
1. No comments, well explain and well written.Conclusions
No comments, well written
No comments, well explain and well written.
Author Response
Thank you for your agreement. We appreciate your informative comments. Bests.
Reviewer 4 Report (New Reviewer)
The authors have revised their paper as per all my comments and feedback. I do not have any additional comments at this point. I recommend the publication of the paper in its current form.
Author Response
Thank you for your great support. Bests.
Reviewer 5 Report (New Reviewer)
The authors successfully addressed my comments. In my opinion, the paper can be considered for publication.
Author Response
We are grateful for your big support. Bests.
This manuscript is a resubmission of an earlier submission. The following is a list of the peer review reports and author responses from that submission.
Round 1
Author Response
Thanks to all reviewers for their informative and constructive comments. We checked all the comments and restructured the manuscript entirely. Please review the R2 version. thank you for your contribution again.
Reviewer 2 Report
How can social media, specifically Twitter, be used to monitor public opinion on biodiversity and human-nature interactions?
What has been the extent of research conducted using Twitter in relation to climate change and biodiversity conservation?
In what ways can the monitoring of social media data, including Twitter data, aid in achieving Aichi Target 1 for biodiversity conservation?
What are the direct and indirect drivers of global change in nature according to the framework proposed by the Intergovernmental Science-Policy Platform on Biodiversity and Ecosystem Services (IPBES)?
How can transformative change be achieved in order to create a more sustainable future, and what are some of the key leverage points and priority governance interventions that can be used to bring about this change?
How were hashtags extracted from tweets related to biodiversity, and what preprocessing techniques were used?
What is the Shannon entropy, and how was it used to evaluate the diversity index of interest in biodiversity?
What methods were used to analyze the interest in biodiversity among different language communities, and what were the results of the analysis?
How were hashtags used in the analysis of biodiversity interest among different language communities, and what insights were gained from this approach?
How do different language communities vary in their use of hashtags and engagement with tweets about biodiversity?
What trends can be observed in the time series of the number of tweets and engagements by language communities in relation to biodiversity?
Author Response

(The authors gave the same response as above.)

Reviewer 3 Report
This article attempts to illustrate the process of identifying commonalities across seven language communities by comparing hashtags in Tweets including the term biodiversity and determining differences in interest and concern for biodiversity from past to present.
The novelty of the paper is not clear. The authors in 2021 collected data from 2006 through 2021 to assess past and present differences in interest and concerns about biodiversity. The data time span is too long for a single collection and biases the investigation. Social networks are dynamic by nature. Users add, change, delete content. Thus, many of the older tweets were deleted and the language used was changed. So the findings could not be validated using approaches such as natural language processing of user tweets. Therefore, the current methodology of the article is limited, and the findings are not new. As a result, I recommend rejecting the article in its current form.
Author Response

(The authors gave the same response as above.)

Reviewer 4 Report
The paper entitled" Twitter mining for detecting interest trends on Biodiversity: messages from seven language communities " studies the mining of text in tweets.
I found the article interesting, insightful, and particularly well-written. I think the authors have tried to put their findings into context. The authors, compares the biodiversity-related hashtags of seven language communities and shows the uniqueness of the Japanese community. It also examines the changes in interest and concern about biodiversity over time.
Specific Comments:
1--There are many errors in the text and the English language needs improvement.
2-- The writing is sloppy in many places.
3--The analysis of the previous works is not satisfying. The authors should highlight more clearly the main pros and cons of their approach. In Section 2, all important recent works should be properly reviewed and compared with each other. All the formulas are already presented in the this part. What was your purpose without any proper analysis?
4--There are also some other relevant papers which are not cited in this manuscript.
5-- Introduction section incomplete. It needs to be contributions highlighted. Also, different with other methods is not satisfying/clear.
6-- Contribution is weak. Only question is provided, which is less informative.
7-- Proposed method section needs to be improved. Pseudo code should be include in a table, and inputs and outputs are highlighted.
8--The section of results is poorly expressed and the way of comparisons, etc. is not clear.
9--The resolution is too low (I,e Figure 8). The caption of the figure must be abjusted to the Figure. Adding some detail that clearly describes the author's figure and explains the figures is also suggested.
10--Can the author provide each language instance sample.
11. Which language the author focused. Its hard to understand, the author mentioned 7 languages (including Japanese). What is the purpose of comparing these language? I recommend to provide comparision evaluation performance (results) with previous work.
Author Response

(The authors gave the same response as above.)

Round 2
Reviewer 3 Report
I stand by my decision to reject the manuscript. The time-frame for data collection in the study, spanning from 2006 to 2021, raises concerns about the validity and reliability of the data.
Thank you
Author Response
We appreciate the reviewer’s suggestion very much. As to the reviewer's concern, we re-emphasized the time period bias of the tweet collection process and a potential approach to avoid it. Thank you again.